⊙ | **Open Peer Review** | Mycology | Research Article

# Argonaute and Dicer are essential for communication between *Trichoderma atroviride* and fungal hosts during mycoparasitism

Eli Efrain Enriquez-Felix,[1] Camilo Pérez-Salazar,[1] José Guillermo Rico-Ruiz,[1] Ana Calheiros de Carvalho,[2] Pablo Cruz-Morales,[2,3] José Manuel Villalobos-Escobedo,[1,3,4,5] Alfredo Herrera-Estrella[1,3]

**ABSTRACT** *Trichoderma* species are known for their mycoparasitic activity against phytopathogenic fungi that cause significant economic losses in agriculture. During mycoparasitism, *Trichoderma* spp. recognize molecules produced by the host fungus and release secondary metabolites and hydrolytic enzymes to kill and degrade the host's cell wall. Here, we explored the participation of the *Trichoderma atroviride* RNAi machinery in the interaction with six phytopathogenic fungi of economic importance. We determined that both Argonaute-3 and Dicer-2 play an essential role during mycoparasitism. Using an RNA-Seq approach, we identified that perception, detox, and cell wall degradation depend on the *T. atroviride*-RNAi when interacting with *Alternaria alternata, Rhizoctonia solani* AG2, and *R. solani* AG5. Furthermore, we constructed a gene co-expression network that provides evidence of two gene modules regulated by RNAi, which play crucial roles in essential processes during mycoparasitism. In addition, based on small RNA-seq, we conclude that siRNAs regulate amino acid and carbon metabolism and communication during the *Trichoderma*-host interaction. Interestingly, our data suggest that siRNAs might regulate allorecognition (*het*) and transport genes in a cross-species manner. Thus, these results reveal a fine-tuned regulation in *T. atroviride* dependent on siRNAs that is essential during the biocontrol of phytopathogenic fungi, showing a greater complexity of this process than previously established.

**IMPORTANCE** There is an increasing need for plant disease control without chemical pesticides to avoid environmental pollution and resistance, and the health risks associated with the application of pesticides are increasing. Employing *Trichoderma* species in agriculture to control fungal diseases is an alternative plant protection strategy that overcomes these issues without utilizing chemical fungicides. Therefore, understanding the biocontrol mechanisms used by *Trichoderma* species to antagonize other fungi is critical. Although there has been extensive research about the mechanisms involved in the mycoparasitic capability of *Trichoderma* species, there are still unsolved questions related to how *Trichoderma* regulates recognition, attack, and defense mechanisms during interaction with a fungal host. In this work, we report that the Argonaute and Dicer components of the RNAi machinery and the small RNAs they process are essential for gene regulation during mycoparasitism by *Trichoderma atroviride*.

**KEYWORDS** sRNA, biocontrol, RNAi, transcriptome, gene expression network, secondary metabolites

**Ad Hoc Peer Reviewer** Nicole Donofrio

Address correspondence to José Manuel Villalobos-Escobedo, jose.villalobos@berkeley.edu, or Alfredo Herrera-Estrella, alfredo.herrera@cinvestav.mx.

The authors declare no conflict of interest.

See the funding table on p. 22.

*T*richoderma atroviride is a filamentous fungus widely distributed in nature, usually enriched in soil and plant roots and foliage (1). *T. atroviride* benefits plants by increasing their tolerance to abiotic stress, promoting their vigor and growth, and activating their local and systemic defenses (2, 3). *T. atroviride* is considered a biological

control agent mainly due to its mycoparasitic activity, linked to the production of hydrolytic enzymes, and its capacity to produce a wide range of volatile and non-volatile secondary metabolites (4–7).

Three decades ago, a wave of work on mycoparasitism by *Trichoderma* spp. helped characterize several genes relevant to this process. Those studies identified genes involved in mycoparasitism, like *prb1*, encoding a proteinase induced when confronted with *Rhizoctonia solani*, causing growth inhibition in the phytopathogen (7, 8). Similarly, *ech42* and *nag1*, encoding an endochitinase and an N-acetyl-β-D-glucosaminidase that participate during mycoparasitism by *T. atroviride* (7, 9, 10), are induced even before contact with a host fungus, suggesting that *Trichoderma* spp perceives diffusible molecules from the host fungus, likely chito-oligosaccharides, or N-acetyl glucosamine (7, 11, 12).

Seidl and collaborators in 2009 reported that *T. atroviride* co-cultured with *Botrytis cinerea* and *R. solani* overexpressed genes enriched in amino acid metabolism and post-translational processing, as well as genes that code for different heat shock proteins, tRNA synthetases, proteases, and signal transduction related proteins (13). Furthermore, with the genome sequencing of *T. virens, T. reesei,* and *T. atroviride* (14), genes with possible relevance for mycoparasitism were reported, among them those encoding glycoside hydrolases and carbohydrate-binding domain proteins and genes related to secondary metabolite production such as non-ribosomal peptide synthases (NRPS) and polyketide synthases (PKS). Despite the extensive studies on the antagonistic capacity of *T. atroviride,* we know very little about the mechanism of perception and transcriptional regulation involved in response to a specific fungal host.

RNA interference (RNAi) is a highly conserved mechanism in eukaryotic organisms. This mechanism regulates gene expression through small RNAs (sRNAs) of approximately 20–30 nucleotides (15). sRNAs play an essential role in gene silencing at the transcriptional and post-transcriptional levels (16). The core components of RNAi are Dicer, an RNase III that processes long double-stranded RNA (dsRNA) into 21–26 nucleotide RNA fragments, and RNA-dependent RNA polymerases that convert aberrant RNA into dsRNA that Dicer proteins can process. The dsRNA fragments produced by Dicer are loaded in Argonaute proteins, which form part of the RNA-induced silencing complex (RISC) (17). In *Neurospora crassa, Fusarium graminearum, Mucor circinelloides, T. atroviride, Clonostachys rosea, Magnaporthe oryzae, Verticillium dahliae, Metarhizium robertsii,* and *Sordaria macrospora*, mutants in the RNAi machinery present morphological and reproductive alterations, suggesting that these components regulate vegetative processes, hyphal development, and spore production (18–27). In other fungi, such as *B. cinerea, R. solani,* and *Beauveria bassiana,* the RNAi machinery is involved in processes related to pathogenesis (28–30). Therefore, we wondered whether small RNAs could exert part of this complex gene regulation. Furthermore, the relevance of small RNAs in mycoparasitism has not been studied so far in *T. atroviride*. The *T. atroviride* genome encodes two Dicer (Dcr), three Argonaute (Ago), and three RNA-dependent RNA polymerases (Rdr). Previously, we reported that Dcr2 and Rdr3 are involved in the regulation of asexual development and that Dcr2 processes most of the sRNAs of 21–22 nucleotides, which are the most abundant sRNAs in *T. atroviride* (21).

Here, we thoroughly describe the role of the RNAi machinery and sRNAs during mycoparasitism. Screening with mutants in all components of the RNAi machinery and the WT strain of *T. atroviride* in confrontation with six phytopathogenic fungi indicated that this machinery plays a significant role in mycoparasitism. Furthermore, RNA sequencing led us to determine that the transcriptional landscape of *Trichoderma* during the interaction is host-specific and significantly controlled by Dcr2 and Ago3. A gene co-expression network analysis uncovered two main modules governed by RNAi regulating the processes of degradation and, host perception and degradation. Finally, small RNA sequencing allowed us to identify sRNAs potentially involved in regulating a *Trichoderma*-pathogenic fungus mycoparasitic interaction. The predicted targets of the *T. atroviride* small RNAs are involved in genetic information processing, carbohydrate

metabolism, signaling, transport, and amino acid metabolism. Interestingly, we found highly induced siRNAs and a milRNA in the WT strain absent in the Δdcr2 mutant, which could regulate several *het* and transport genes of *A. alternata*. Our results indicate that in *T. atroviride,* sRNAs play an important role during the molecular dialogue between fungal species.

## RESULTS

### Dicer-2 and Argonaute-3 are required to inhibit the growth of phytopathogenic fungi

To evaluate the biocontrol capacity of the *T. atroviride* WT and RNAi mutant strains, we performed dual confrontation assays against *Alternaria alternata, Fusarium oxysporum, Fusarium* spp., *B. cinerea*, and two strains of *R. solani*, one belonging to anastomosis group 2 (AG2) and one to group 5 (AG5) (Table 1). Even though the WT strain overgrew all fungi (Fig. S1), it took longer to overgrow *A. alternata* than *R. solani* AG5, while it overgrew faster *R. solani* AG2 than AG5. By contrast, the Δdcr2 mutant and the double Δdcr1Δdcr2 mutant of *T. atroviride* showed reduced capacity to overgrow *A. alternata, R. solani* AG2, *R. solani* AG5, and *B. cinerea* (Fig. 1A; Fig. S1). We observed a clear zone of growth inhibition of the *Trichoderma* mutants by the fungal hosts (dashed rectangles in Fig. 1A and B). By contrast, the mutant strains in any RNA-dependent RNA polymerases (Δrdr1, Δrdr2, and Δrdr3) and those affected in *ago1* and *ago2* overgrew the fungal hosts like the WT (Fig. S1). Interestingly, the Δdcr2 and Δago3 strains did not stop the growth *of A. alternata* and *R. solani* AG5 and did not overgrow them (Fig. 1A and B).

Due to the significance of antibiosis in biocontrol mediated by *Trichoderma*, we evaluated the impact of diffusible compounds released by the RNAi mutant strains on the growth of various fungal species (Fig. 1C; Fig. S2). Briefly, we grew the different *T. atroviride* strains on PDA plates covered by a cellophane sheet to allow the remotion of *Trichoderma* and diffusion of metabolites, and then we inoculated the indicated fungal hosts. All mutant strains inhibited the growth of the different fungal hosts to a similar degree to that observed for the WT, except for Δrdr2, which showed a strongly reduced capacity to inhibit the growth of all tested hosts (Fig. S2). Conversely, the effect of diffusible compounds from a fungal host (*R. solani* AG5 and *A. alternata*) on the growth of the RNAi mutant strains was evaluated (Fig. 1D and E; Fig. S3 and S4). Interestingly, the Δdcr2 and the Δdcr1Δdcr2 double mutant showed a significant growth reduction compared to the WT (Fig. 1D and E). These results suggest that the Dcr2 component regulates genes involved in detoxifying diffusible metabolites.

**TABLE 1** Fungal strains used in this study

| Strains | Origin (reference) |
| --- | --- |
| *T. atroviride* WT | IMI206040 (International Mycological Institute) |
| *T. atroviride* Δdcr1 | Derived from *T. atroviride* IMI206040 (21) |
| *T. atroviride* Δdcr2 | Derived from *T. atroviride* IMI206040 (21) |
| *T. atroviride* Δdcr1/Δdcr2 | Derived from *T. atroviride* IMI206040 (21) |
| *T. atroviride* Δago1 | Derived from *T. atroviride* IMI206040 (21) |
| *T. atroviride* Δago2 | Derived from *T. atroviride* IMI206040 (21) |
| *T. atroviride* Δago3 | Derived from *T. atroviride* IMI206040 (21) |
| *T. atroviride* Δrdr1 | Derived from *T. atroviride* IMI206040 (21) |
| *T. atroviride* Δrdr2 | Derived from *T. atroviride* IMI206040 (21) |
| *T. atroviride* Δrdr3 | Derived from *T. atroviride* IMI206040 (21) |
| *Botrytis cinerea* | This work, isolated from infected strawberries |
| *Fusarium oxysporum* | This work, isolated from tomato roots |
| *Rhizoctonia solani* AG2 | Kindly donated by Dr. June Simpson (CINVESTAV-Irapuato), isolated from potato tubers |
| *Rhizoctonia solani* AG5 | Kindly donated by Dr. June Simpson (CINVESTAV-Irapuato), isolated from potato tubers |
| *Alternaria alternata* | This work, isolated from tomato leaves |

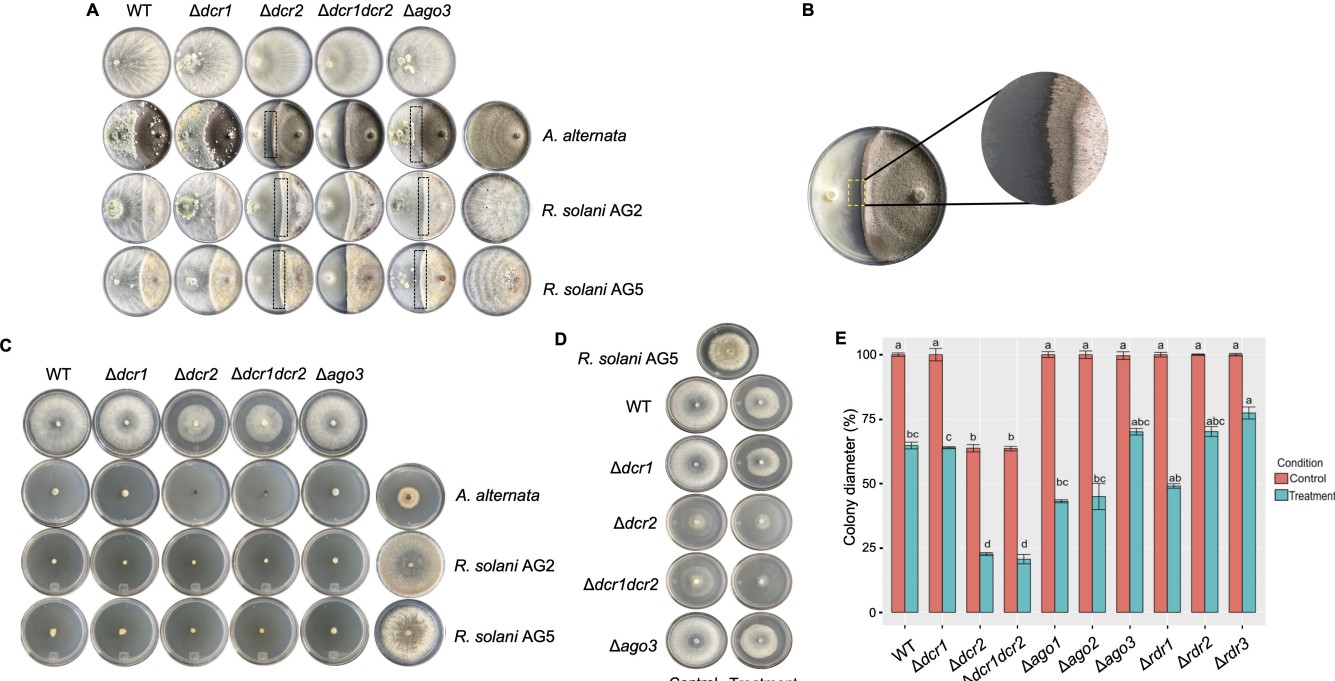

FIG 1 The mycoparasitic ability of the wild-type (WT) strain and RNAi mutant strains. (A) Dual culture assays of RNAi mutant strains against some fungal hosts. The *T. atroviride* strains were inoculated on the left side, and the fungal hosts on the right hand. The plates in the top row show the *T. atroviride* strains growing alone. The plates in the rightmost column correspond to the fungal hosts growing alone. The rectangles indicate the interaction area between the two fungi where no contact was observed. (B) Closeup of the confrontation between *Δdcr2* strain vs *A. alternata* showing a clear growth inhibition zone. (C) Representative photographs showing the effect of the diffusible compounds of the *T. atroviride* RNAi mutant strains on different fungi. (D) Representative photographs showing the effect of *R. solani* AG5 diffusible compounds on the *T. atroviride* RNAi mutant strains. The left column represents the control strains without the diffusible compounds of *R. solani* AG5, while the right column shows the treatment with *R. solani* AG5 diffusible compounds. (E) Colony diameter of *T. atroviride* RNAi mutant strains growing in the presence of *R. solani* AG5 diffusible compounds (blue bars) or without them (red bars). The values represent the mean of three biological replicas performed at different times. A one-way ANOVA and a Tukey test were performed to determine statistical differences among the strains in the same treatment ($P < 0.05$).

## *Trichoderma atroviride* exhibits host-specific transcriptional responses during mycoparasitism

To identify the genes regulated by Dcr2 and Ago3 during mycoparasitism, we constructed RNA-Seq libraries from the *T. atroviride* WT, *Δdcr2,* and *Δago3* strains in confrontation with *A. alternata*, *R. solani* AG2, and *R. solani* AG5 as hosts due to the phenotypes observed against them. The different interactions were evaluated at three stages: before contact (BC), during contact (DC), and after contact (AC) with the host fungus. The number of differentially expressed genes (DEGs) at every stage and interaction is summarized in Fig. 2 and detailed in Data Set 1. A multidimensional scaling (MDS) plot analysis showed that each of the replicates of the WT strain libraries clustered together, close to the *Δago3* libraries. Whereas the libraries of the *Δdcr2* strain clustered together and away from the *Δago3* and WT libraries (Fig. 2A).

The transcriptomic response of the WT was host- and stage-specific. In the *T. atroviride* confrontation with *A. alternata*, we found more DEGs, particularly at the AC stage; however, at the BC and DC stages, the *R. solani* AG2 interaction showed the highest number of DEGs (Fig. 2B).

At the BC stage, *T. atroviride* showed an up-regulation of genes enriched in translation, RNA metabolism, and amino acid metabolism only with *R. solani* AG2 (Fig. 3, cluster IX). However, genes enriched in translation, protein folding, and amide biosynthesis were down-regulated at the AC stages in the *T. atroviride* confrontation with *R. solani* AG5 and *A. alternata* (Fig. 3, cluster X). Particularly in the *A. alternata* confrontation, *T. atroviride*

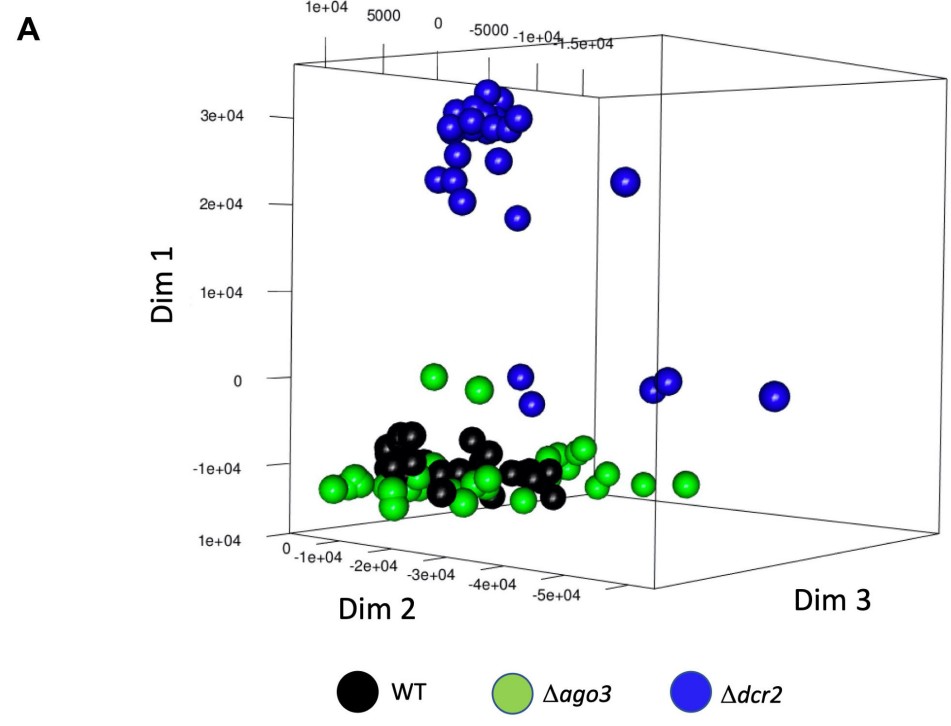

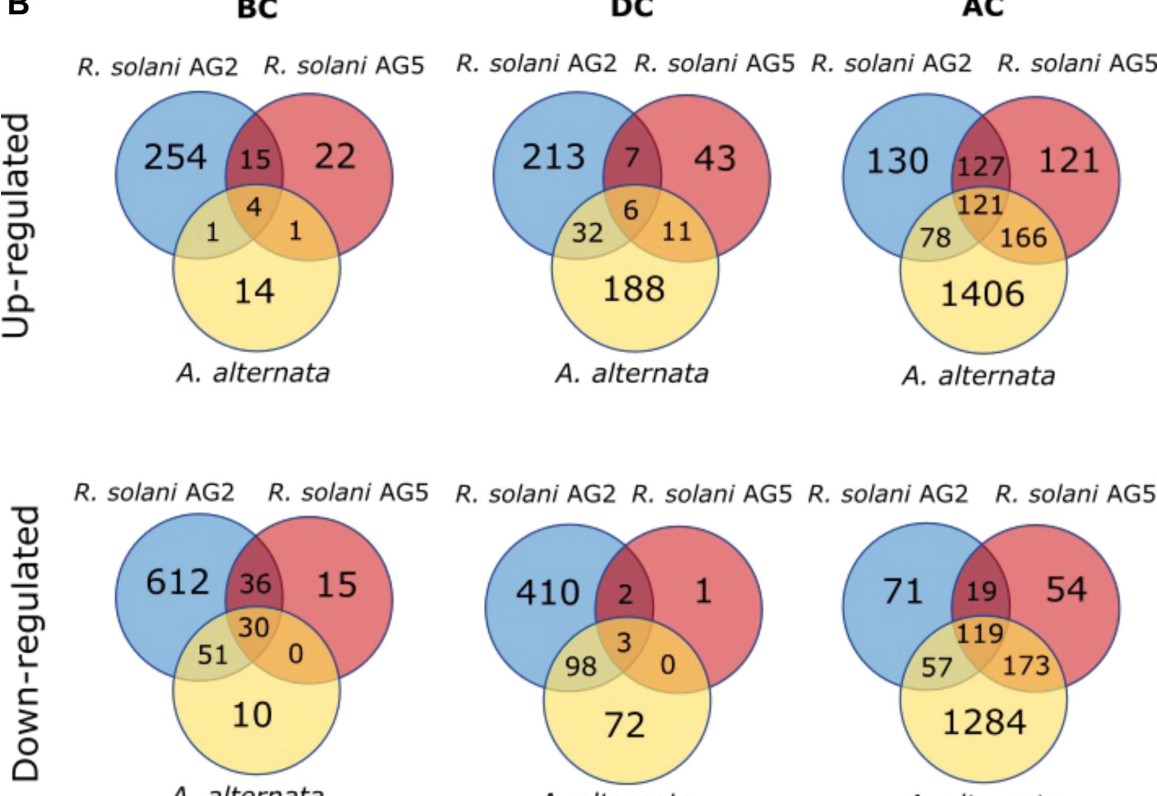

**FIG 2** Global analysis of the RNA-seq experiment during mycoparasitism. (A) Distribution of the counts per million (MDS plot) in the WT, Δdcr2, and Δago3 libraries of *T. atroviride*, facing *A. alternata*, *R. solani* AG2, and *R. solani* AG5. (B) Venn diagram classifying DEGs, including an adjusted FDR q-value (FDR < 0.05) in the *T. atroviride* WT strain compared against the control condition. Before contact (BC), during contact (DC), and after contact (AC).

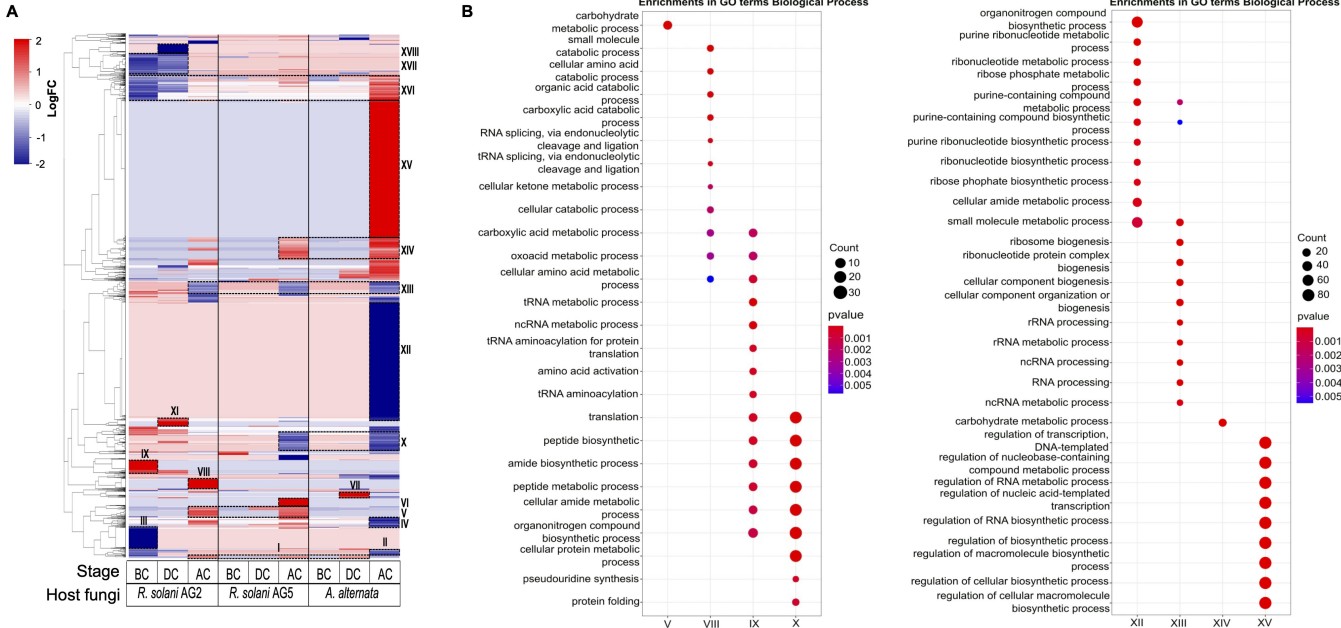

**FIG 3** Differentially expressed genes and gene ontology (GO) functional categories from *T. atroviride* (WT) interactions. (A) The figure shows a heatmap of DEGs for all confrontations and stages BC, DC, and AC. Roman numerals represent the clusters formed by Pearson correlation. The different clusters identified in the dendrogram are represented by the colored bar at the left of the heatmap. (B) GO enrichment (Biological processes) of genes in clusters V, VIII, IX, X, XII, XIII, XIV, and XV (Right). The genes in these clusters are listed in Data Set 2.1.

showed down-regulated genes enriched in RNA metabolism at the AC stage while at the same time showed up-regulated genes involved in transcription regulation, RNA biosynthesis, and macromolecule biosynthesis regulation (Fig. 3, clusters XII and XV). This result contrasts with the response observed in the confrontation with *R. solani* AG2 at the AC stage, where genes enriched in RNA metabolism were up-regulated (Fig. 3, cluster VIII). In summary, *T. atroviride* showed an up-regulation in protein and amino acid biosynthesis in the BC stage; however, when *T. atroviride* was in contact and overgrowing the host (DC and AC stages), it shifted to the expression of catabolism related genes.

In addition, we observed that the number of transporter genes was higher in the confrontation of *T. atroviride* with *A. alternata* than when confronted with *R. solani* AG5/AG2, particularly at the AC stage. In the first case, 81 genes coding for different types of transporters (MFS and ABC) were up-regulated compared to 32 and 15 in the other confrontations. Following the same pattern, we observed more up-regulated genes coding for CAZymes in the *T. atroviride-A. alternata* confrontation (72 genes, Fig. S5; Data set 2.2) than in confrontation with *R. solani* AG5/AG2 (32 genes, Fig. S5; Data set 2.2). We also observed that CAZymes (23 genes) were activated only against the *R. solani* AG5/AG2 strains, such as polysaccharide lyases (PLs) (clusters II, VIII, and IX, Fig. S5). Furthermore, we found host-specific genes encoding PKS and NRPS biosynthetic gene clusters that responded only to the *R. solani* AG5/AG2 strains (Fig. 3, clusters V, IV, and VIII) and other NRPSs specific to the interaction with *A. alternata* (Fig. 3, cluster XV).

In summary, *T. atroviride* could suppress several metabolic processes, depending on the stage (BC, DC, or AC) and the fungal host confronted. Based on our observations during the AC stage interaction with *A. alternata,* we propose that a robust post-tran-scriptional regulation occurs in *T. atroviride* at this stage of the interaction (Fig. 3).

## Dcr2 regulates transmembrane transport and protein and carbohydrate metabolism

To identify the genes regulated by Dcr2 and Ago3, we conducted a differential expression analysis comparing the Δ*dcr2* and Δ*ago3* mutant strains to the WT strain at each stage of the interaction. In the Δ*dcr2* vs WT comparison during the interaction with *A. alternata* (Fig. 4A), we observed the highest number of DEGs, with 911 genes up-regulated and 645 genes down-regulated at the BC stage. At the DC stage, we observed 365 up-regulated genes and 65 down-regulated genes, while at the AC stage, we observed 697 up-regulated genes and 627 down-regulated genes. Interestingly, in the Δ*ago3* vs WT comparison during the same stage (AC), we also observed a substantial number of differentially regulated genes in the interaction with *A. alternata*, with 813 genes up-regulated and 641 genes down-regulated.

The Δ*dcr2* strain transcriptomic response changed depending on the stage; at the BC stage, we detected up-regulated genes enriched in carbohydrate metabolism, fatty acid biosynthesis, and amino acid transport (Fig. 5A), while the down-regulated genes were related to RNA and tRNA splicing, phosphorylation, and phosphorelay signal transduction (Fig. 5B). Such responses were similar at the DC stage. However, genes related to carboxylic acid anabolism and biosynthesis were induced after contact. By contrast, groups of genes enriched in transcription regulation and RNA metabolism were down-regulated (Fig. 5B). We also observed that genes coding for CWDEs and other CAZymes were affected particularly with *A. alternata* (Fig. S6; Data set 2.3). Some genes relevant to mycoparasitism such as chitinases and glucanases were down-regulated mainly in the AC stage. This down-regulation contrasts with what was observed for the WT strain, which showed up-regulation of genes encoding CWDEs at the AC stage.

Due to the inhibition zone observed in the confrontations between Δ*dcr2* and the fungal hosts (Fig. 1) and the GO terms associated with transport found in the transcriptomic data, we also searched for genes encoding MFS (Major Facilitator Superfamily) and

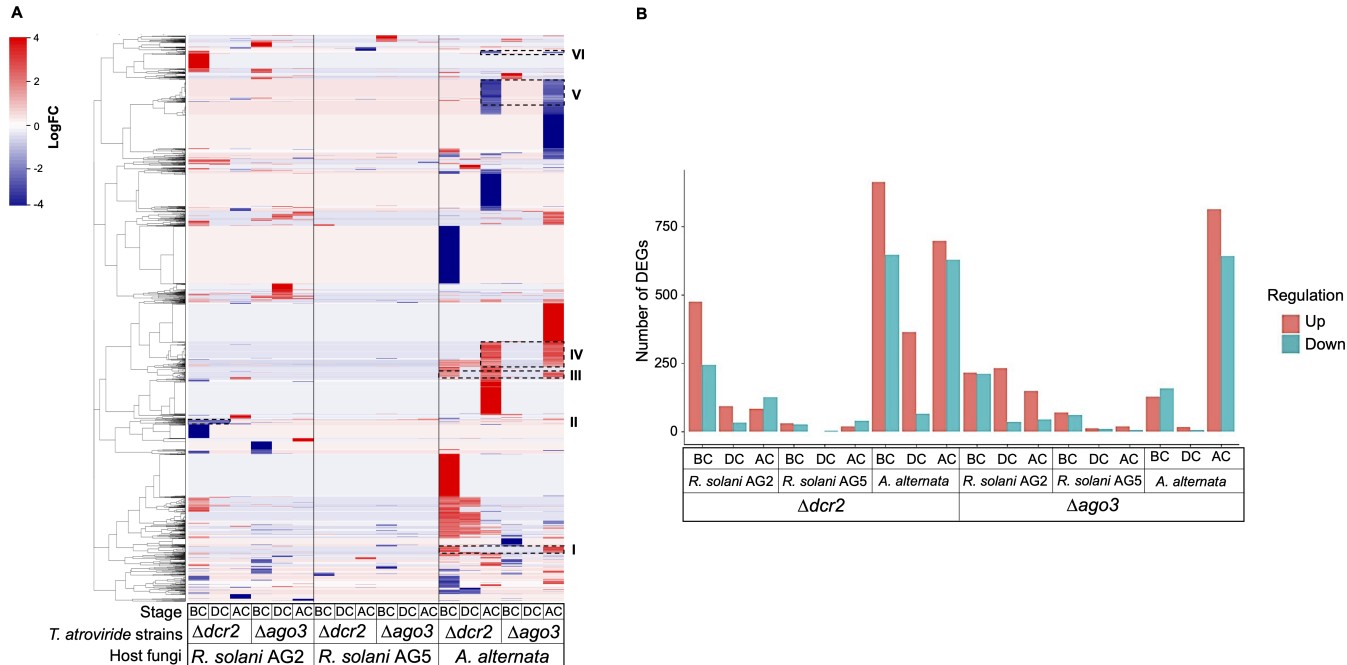

**FIG 4** DEGs of RNAi mutant strains in confrontation with *R. solani* AG2/AG5 and *A. alternata*. (A) Heatmap showing the DEGs in Δ*dcr2* and Δ*ago3* mutant strains (*P* < 0.05) relative to the WT strain at three different stages of mycoparasitism (BC, DC, and AC). The black delimited rectangles are gene clusters shared between mutants. The different clusters identified in the dendrogram are represented by the colored bar at the left of the heatmap. The gene clustering was performed by correlation using the Pearson method. The GO terms found in the clusters are detailed in Data Set 2.5. (B) The graph shows the number of differentially expressed genes DEGs in Δ*dcr*2 and Δ*ago*3 compared to the WT strain. The up-regulated and down-regulated genes are represented by red and blue bars, respectively.

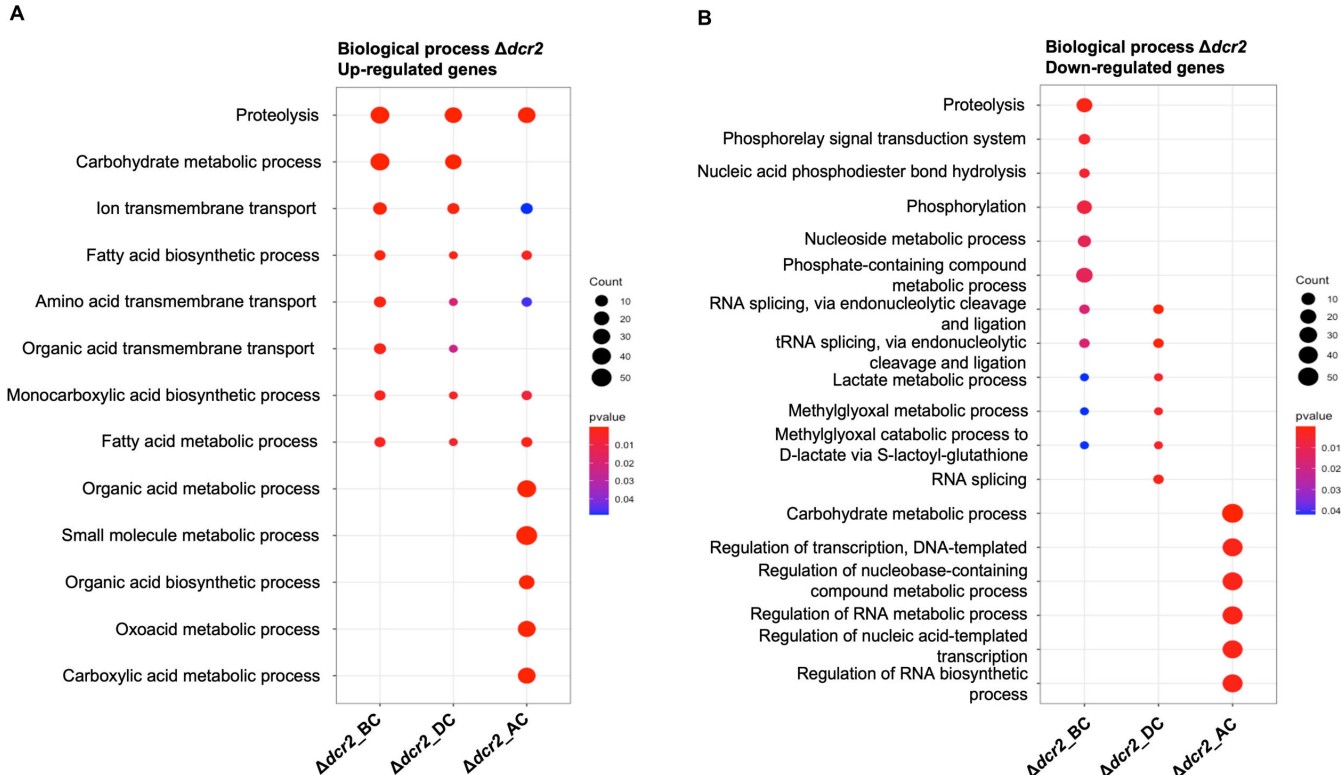

**FIG 5** Gene ontology (GO) enrichment analysis of DEGs. (A) Plot showing the GO terms of up-regulated genes of Δ*dcr*2 mutant against *A. alternata* at the three stages: BC, DC, and AC. (B) Plot showing the GO terms of down-regulated genes of Δ*dcr*2 mutant against *A. alternata* at the three stages: BC, DC, and AC. ($P <$ 0.05).

ABC transporters in the Δ*dcr2* strain libraries. As a result, we observed multiple transporters repressed at the BC and AC stages in the Δ*dcr2*-*R. solani* AG2 and Δ*dcr2*-*A. alternata* confrontations (Fig. S7 and S8; Data set 2.4). Interestingly, the observed transport misregulation was host-dependent. Most DEGs coding for ABC and MFS transporters in the Δ*dcr2* strain were observed at the AC stage against *A. alternata* (Fig. S7, cluster II, and Fig. S9). However, in the *R. solani* AG2 confrontation, most down-regulated genes coding for transporters were found at the BC stage (Fig. S7, cluster VIII). The observed misregulation of transport-related genes could explain the incapacity of Δ*dcr2* to detoxify the metabolites released by the *A. alternata* and *R. solani* AG5/AG2 strains and the growth inhibition observed in the dual culture assays (Fig. 1A and B).

Finally, due to the relevance of secondary metabolism in mycoparasitism (4, 14), we performed a prediction of biosynthetic gene clusters in *T. atroviride* to search for genes whose expression may be affected in the Δ*dcr2* strain. We identified core biosynthetic genes such as PKS and NRPS in 33 biosynthetic gene clusters (BGC). We observed evident defects in the expression of three PKSs and three NRPS belonging to three different BGCs in the Δ*dcr2* strain. These genes were expressed in the wild-type strain in all interactions but repressed in the Δ*dcr2* mutant (Fig. 6). Remarkably, within BGC 7.1, we found a gene (Tatro_010797 gene) with a domain organization like that previously described in the NRPSs involved in the biosynthesis of the insecticidal hexa-depsipeptides destruxin, isaridin, and isariin (Fig. 7A), all found in Hypocreales of the genus *Beauveria* (31). This gene is repressed in the Δ*dcr2* mutant in both mycoparasitic and control conditions (Fig. 7B).

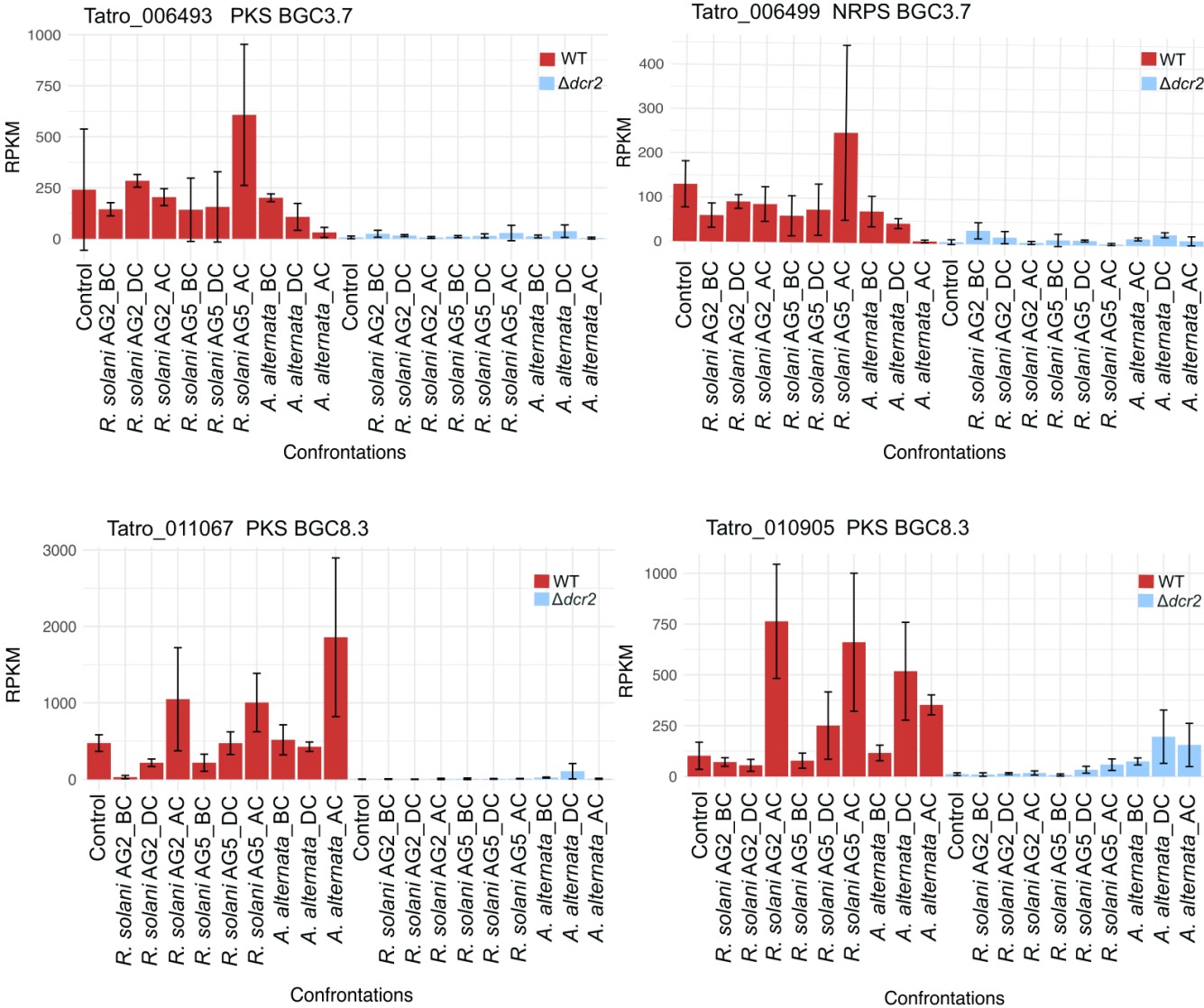

**FIG 6** Expression profiles of secondary metabolism genes in WT and Δ*dcr2* strains. The graph shows the expression level of the indicated gene in the *T. atroviride* WT strain (red bars) and the DΔ*dcr2* mutant (blue bars). The gene id, gene product, and biosynthetic gene cluster (BGC) are indicated in each graph.

## Ago3 is involved in the regulation of translation and host recognition

As highlighted above, in the comparison of Δ*ago3* vs WT, we detected a high number of differentially expressed genes in response to *A. alternata* and *R. solani* AG2, particularly at the BC and AC stages (Fig. 4). At the BC stage, the up-regulated genes in the Δ*ago3* mutant strain were enriched in ribosome biogenesis and fatty acid synthases when confronted with *A. alternata* (Fig. 8A). Besides, in Δ*ago3* we found up-regulated genes encoding effectors such as hydrophobins, CFEM-domain containing genes, and a GLEYA domain or lectin-binding domain-containing gene in all confrontations. These types of effectors have been found expressed in *Trichoderma* species in interaction with plants and mycoparasitism and are proposed to play a role in adhesion (32, 33). By contrast, these genes were down-regulated in all confrontations at the BC and DC stages in the WT, particularly the ones encoding a GLEYA-domain protein and a hydrophobin.

On the other hand, the down-regulated genes in Δ*ago3* were enriched in nucleoside metabolism and amino acid transport. We found genes encoding MFS, transcription factors, heat shock proteins, GTPases, and genes involved in amino acid metabolism (Fig. S10).

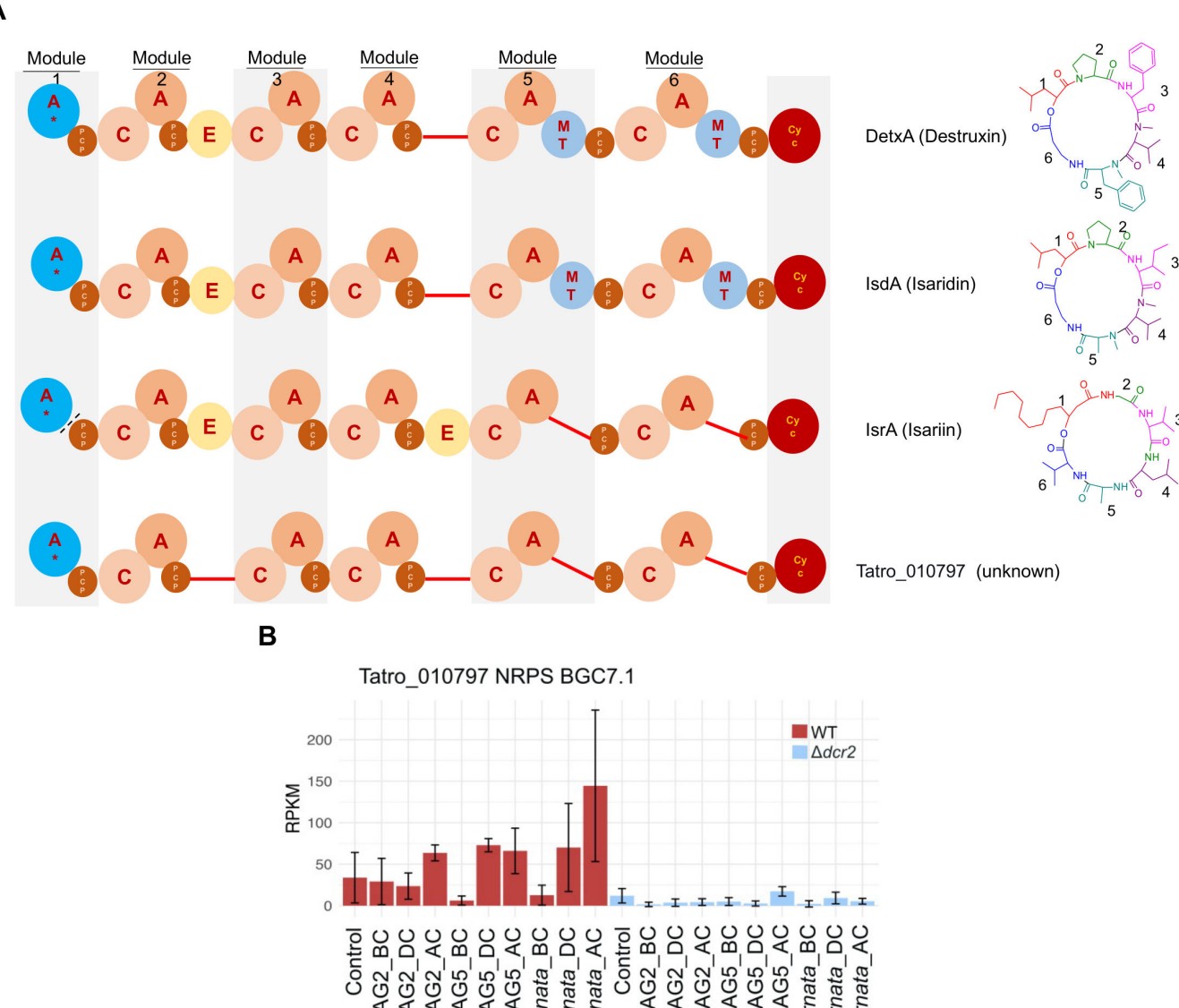

**FIG 7** Biosynthetic prediction and expression analysis of Tatro_010797. (A) The domain organization of DetX, IsdA, and IsrA, the non /ribosomal peptide synthetases involved in the biosynthesis of destruxin, isaridin, and isariin, is represented by circles. Each module incorporates an amino acid; the structures on the right show each amino acid incorporated in the biosynthesis. The synthase encoded by Tatro_010797 is shown at the bottom. The red lines indicate the absence of equivalent domains present in other synthases. A: Adenylation domain A*: adenylation domain specific for a hydroxy acid precursor, PCP: peptidyl carrier protein. C: condensation domain. E: epimerization domain. MT: N-methyl transferase domain. Cyc: Cyclization domain. (B) The graph shows the expression level of Tatro_010797 in the *T. atroviride* WT strain (red bars) and in the D*dcr2* mutant (blue bars).

Although at the DC stage, few DEGs were detected, at the AC stage, we found multiple genes in the *Δago3-A. alternata* interaction (Fig. 4B). The up-regulated genes were enriched in nucleoside metabolism, response to oxidative stress, and carbohydrate metabolism (Fig. 8A). By contrast, the down-regulated genes were enriched in protein biosynthesis and translation (Fig. 8B). These results indicate that both Dcr2 and Ago3 regulate protein metabolism, but Ago3 positively regulates amino acid biosynthesis, particularly at the BC stage.

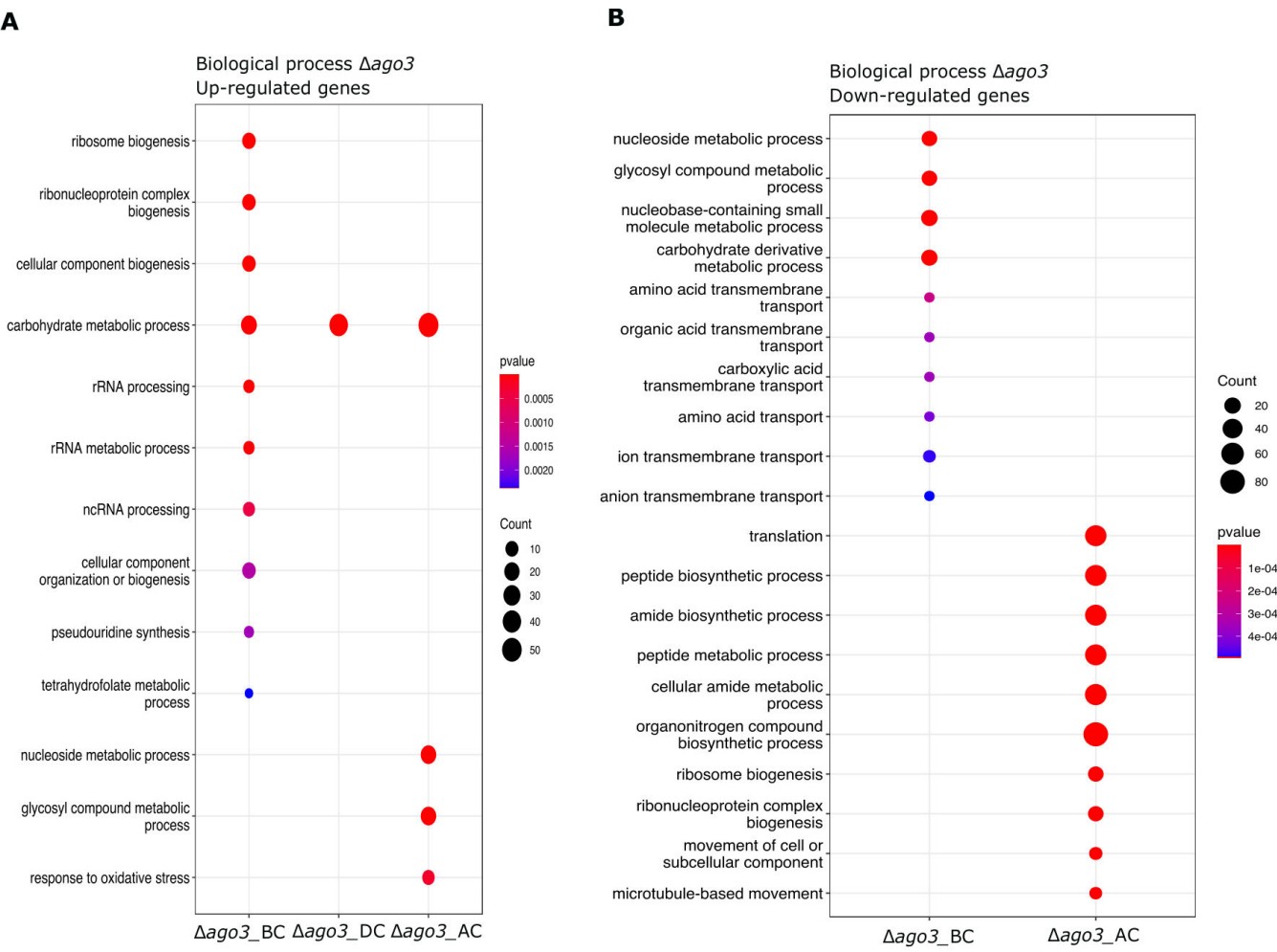

**FIG 8** Gene ontology (GO) enrichment of DEGs in Δago3 strain vs WT strain. (A) Plot showing the GO terms of up-regulated genes of Δago3 mutant against *A. alternata* at the three stages: BC, DC, and AC. (B) Plot showing the GO terms of down-regulated genes of Δago3 mutant against *A. alternata* at the three stages: BC, DC, and AC (*P* < 0.05).

We also observed shared DEGs between the Δdcr2 and Δago3 mutants during the interaction with the host (Fig. 4A; Data set 2.5). We found up-regulated genes (clusters I, III, and IV in Fig. 4A) enriched in transcription regulation, RNA processing, and response to oxidative stress and down-regulated genes (clusters V and VI in Fig. 4A) enriched in phosphorylation, signal transduction, lipid metabolism, and ion transport. Finally, we found genes down-regulated at all stages encompassing redox enzymes and transporters (Table S1), while the up-regulated genes encode transcription factors and redox enzymes. This indicates that Dcr2 and Ago3 co-regulate genes related to transcription regulation, ROS response and transport during mycoparasitism.

## Dcr2 and Ago3 regulate the expression of entire gene modules and their hub genes

Due to the complex transcriptomic response observed in the WT strain during the confrontations against the different fungal hosts, we constructed a Gene Co-expression Network (GCN) to find modules that could change during the Δdcr2 and Δago3 interactions compared to the WT strain (Fig. 9A).

The WT strain modules enriched in carbohydrate metabolism, transport, and hydrolytic activity (Table S2) increased their expression gradually (pink and brown modules in Fig. 9B). The hub genes of these modules encode an exoglucanase CBH2

**A**

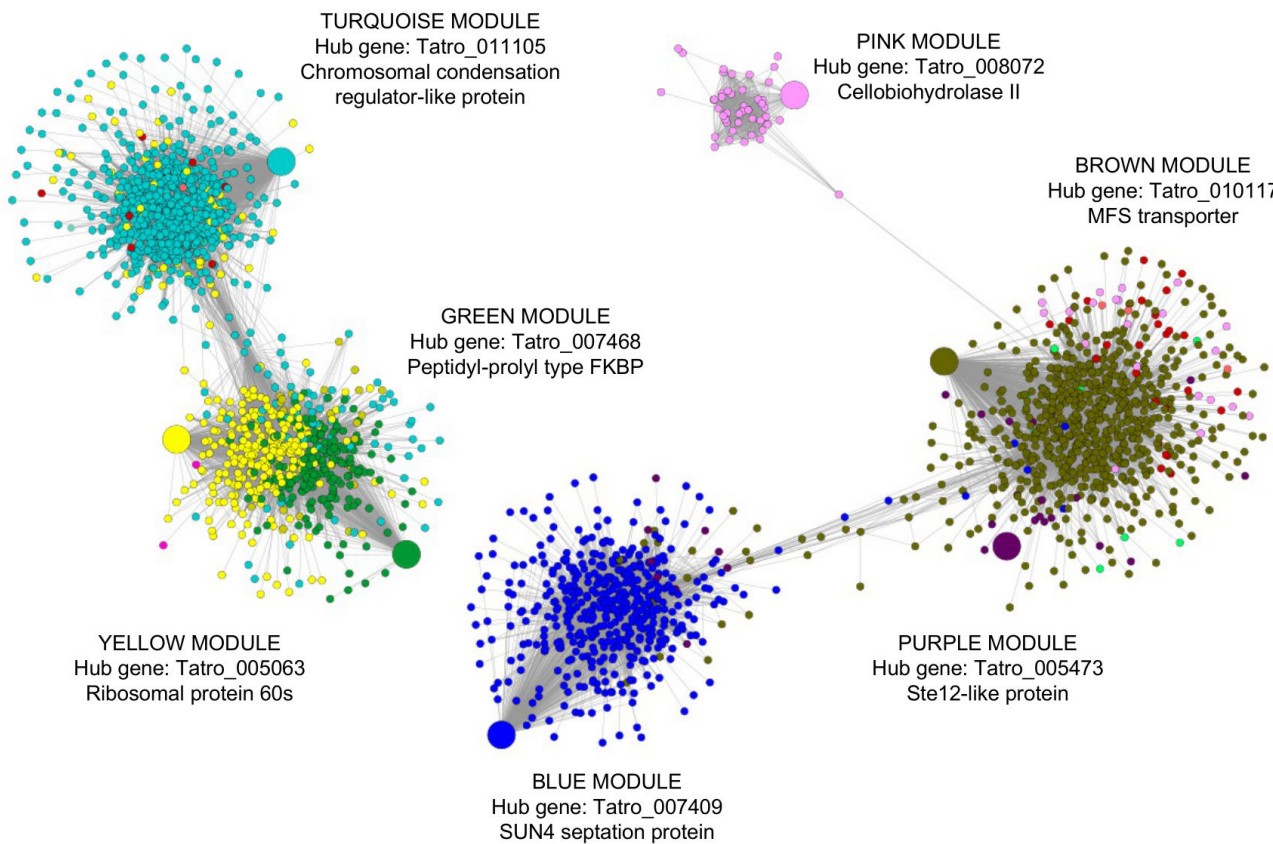

**B**

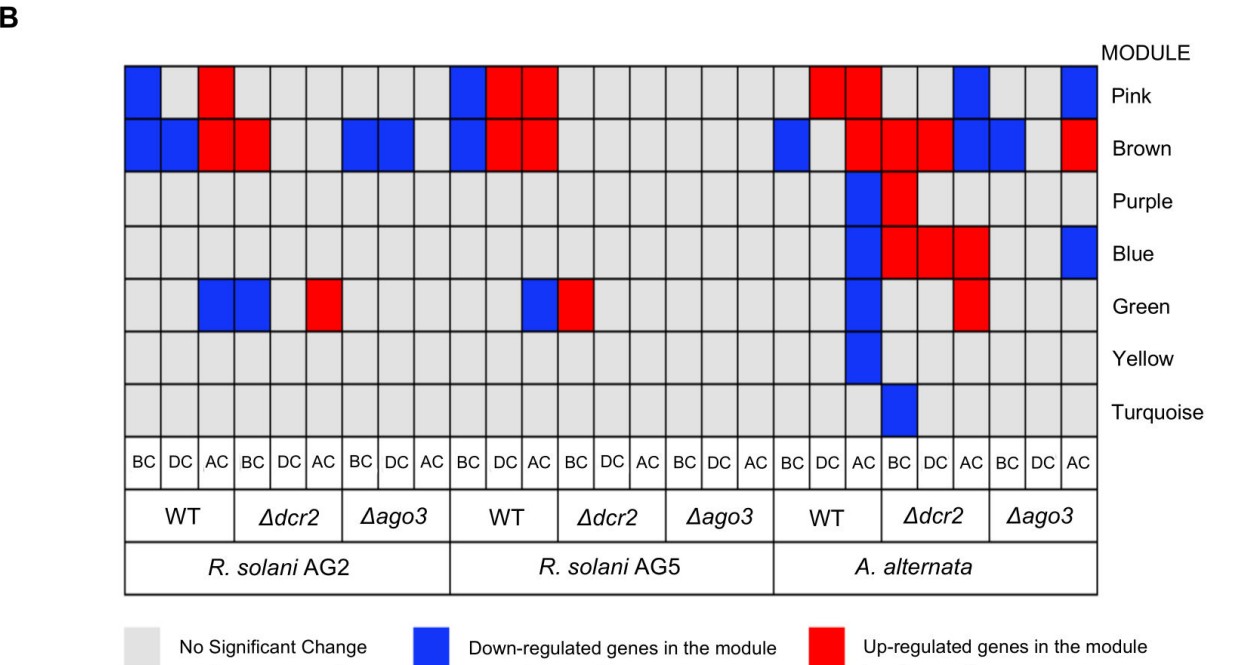

FIG 9 Weighted Gene Co-expression Network of the transcriptomic data from WT and RNAi mutant strains. (A) The figure shows the gene co-expression network constructed using WGCNA with β = 10. Fifteen modules were identified with a total of 7,326 genes. The colors of each node represent the module to which they belong. The modules with the highest number of genes are turquoise (1,359 genes), blue (1,194 genes), brown (1,182 genes), yellow (560 genes), green (427

**FIG 9** (Continued)

genes), and pink (362 genes). The network was visualized in the Cytoscape using a threshold value of 0.19. The nodes represent each gene, and the edges show the connection based on each gene's correlation value (Pearson) according to its co-expression value. The most statistically significant ($P < 0.05$) GO terms for each module are shown. BP: Biological Process. MF: Molecular Function. The largest circle represents the hub gene of the module. (B) Heatmap of the different modules and their expression during the confrontation against the fungal hosts. The up- or down-regulation ($P < 0.05$) of each module in the Δ*dcr*2 and Δ*ago*3 strains is relative to the WT strain and is indicated by colors (red and blue, respectively).

(pink module) and an MFS transporter (brown module) (Table S3). In the pink module, the genes with high connectivity encode glycosyl hydrolases such as glucosidases, glucanases, and mannosidases (Table S4). Accordingly, we suggest this module is involved in host cell wall degradation. By contrast, in all confrontations, the green module decreased its expression during the last stage. The genes in this module are related to translation, ribosome biogenesis, and RNA processing. Meanwhile, modules (blue, purple, and yellow) enriched in signaling, protein and amino acid biosynthesis, and transcription regulation decreased their expression only in the WT-*A. alternata* interaction at the AC stage.

We compared the DEGs found in the modules in the contrasts of the Δ*dcr*2 and Δ*ago*3 strains confronted with *A. alternata* against their controls growing alone with those found for the WT strain. We observed that the pink, brown, and blue modules were the most affected (Fig. 9B). The pink module that increases gradually in the WT was down-regulated at the AC stage in the *A. alternata* confrontation in both mutant strains. The hub gene of this module was down-regulated in the Δ*dcr*2 strain and was not induced in the Δ*ago*3 to the same degree (LogFC = +3.03) observed in the WT strain (LogFC= +10.03), a drastic change in the expression. Likewise, Δ*dcr*2 also showed a decrease in the brown module, following the opposite pattern of the WT strain in the *A. alternata* confrontation. This module was down-regulated in the *R. solani* AG2 vs Δ*ago*3 confrontation but only at the BC and DC stages. The brown module encompasses genes encoding proteins involved in hydrolysis and transport that should be induced gradually; however, they do not follow the same pattern of induction in the mutants as in the WT, and they are even down-regulated in the *A. alternata* confrontation. The blue module was down-regulated only at the AC stage in the WT-*A. alternata* interaction and was up-regulated at all stages in the Δ*dcr2-A. alternata* interaction. While in Δ*ago*3, we observed more repression than in the WT strain at the AC stage. The hub gene belonging to this module (β-glucosidase, Data set 2.7) was down-regulated in both mutant strains at the AC stage in the confrontation with *A. alternata*. By contrast, in the WT, this hub gene did not change its expression at any stage or confrontation.

In summary, Dcr2 and Ago3 regulate the expression of two entire modules (brown and pink) involved in the degradation of cell walls and transport, showing the most drastic response in the WT strain.

## The small RNAs produced by Dcr2 during mycoparasitism

Based on the strong transcriptomic response of *T. atroviride* observed when confronted with *A. alternata* at all stages of the interaction (Fig. 3 and 4), we prepared small RNA-Seq libraries from *T. atroviride* mycelium obtained from the WT strain—*A. alternata* and Δ*dcr*2 —*A. alternata* confrontations in the same conditions as the RNA-Seq libraries. Approximately 18.7 and 16.6 million clean reads from the WT and Δ*dcr*2 libraries were obtained on average of three replicates, respectively.

The most considerable *T. atroviride* fraction of small RNAs is generated from intergenic regions. The size distribution of the small RNAs from the WT strain showed peaks of 21 and 22 nucleotides (Fig. 10A). However, in Δ*dcr*2, we observed that these peaks disappeared, remaining a peak of 25 nt (Fig. 10B). In the *T. atroviride* WT strain, we identified 30 microRNA-like RNAs (milRNAs) in our libraries using ShortStack and miRDeep2 (Table S4). Some of these milRNAs have already been reported and experimentally validated in *T. atroviride* (34). Because these milRNAs have not been reported in miRbase, we performed a BLAST search for their precursors in the NCBI database.

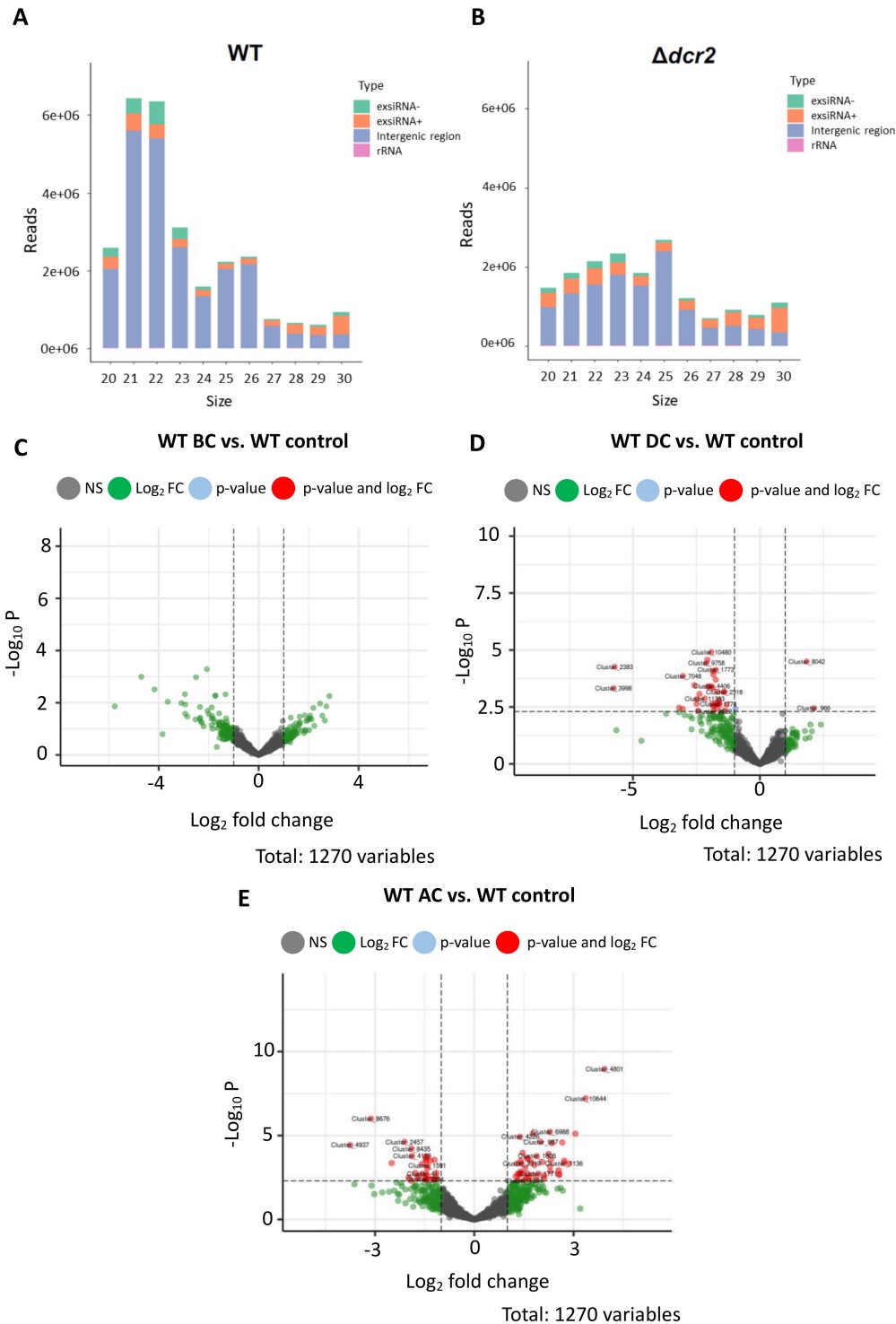

**FIG 10** Small RNA population and differential expression analysis of the *T. atroviride* WT and the Δ*dcr*2 mutant strains. Read distribution of small RNA of 20–30 length. We mapped the small RNA libraries to rRNA, tRNA, snoRNA, snRNA, and ncRNA *T. atroviride* sequences found in the Rfam database (35). We only found reads from rRNA in 20–30 small RNAs. For sRNAs from exons and intergenic zones, the Rsubread package was used. (A) Accumulation profile of small RNAs in the WT strain. (B) Accumulation profile of small RNAs in the Δ*dcr*2 mutant. Differentially expressed small RNAs (DE-sRNAs) of the WT strain at different stages before contact (C), during contact (D), and after contact (E) vs the control growing alone. The red dots represent DE-sRNAs with FDR < 0.05 and log2FC ± 1.

Thirteen milRNAs were shared with other *Trichoderma* species with up to 95% identity (E value <0.05), including *T. asperellum, T. virens, T. reesei, and T. gamsii*, and only one was shared with another genus (*Beauveria*). We also performed a differential expression analysis of the predicted milRNAs. However, only two milRNAs were differentially expressed at the AC stage compared to the control growing alone in the WT strain [Tatro_milR-15 (logFC = 1.89) and Tatro_milR-28 (logFC = −1.24)].

## Mycoparasitism-specific expression of siRNAs

To investigate whether other small RNAs are involved in mycoparasitism, we performed a differential expression analysis of the small RNA clusters (20–26 nt) obtained (Data set 2.6). On average, we detected 2,100 and 789 siRNAs in control and mycoparasitism conditions in the WT strain libraries. Most small RNA clusters were lost in the *Δdcr2* mutant, remaining only 36 and 99 siRNAs in *Δdcr2* (Table S6). We then performed a differential expression analysis of the siRNAs at stages BC, DC, and AC compared to those observed when the *T. atroviride* WT strain grew alone. Differentially expressed clusters of siRNAs (DE-siRNAs) increased according to the stage. No DE-siRNAs were detected before contact; 20 (1 up-regulated and 19 down-regulated) were detected at the DC stage and 75 at the AC stage (45 up-regulated and 30 down-regulated) (Fig. 10C through E; Data set 2.7).

We predicted 1,849 targets of the 95 DE-siRNAs found (Data set 2.8). A gene ontology enrichment analysis of the targets showed that the most frequent enriched terms were primary metabolic process, nitrogen compound metabolic process, macromolecule metabolic process, transport, and transcription (Data set 2.9). In comparison, milRNAs targets are involved in signal transduction and transcriptional and translational regulation (Data set 2.10).

We selected the siRNA's predicted targets up-regulated in *Δdcr2* compared to the WT in our transcriptomic data (Data set 2.11). We found that some targets, including transcription factors, peptidases, and transporters, are indeed overexpressed in *Δdcr2* (Fig. 11; Fig. S11), indicating that the loss of the siRNA in *Δdcr2* affects their expression. We further evaluated the filtered targets using a Wilcoxon test to compare their expression in the *Δdcr2* and *Δago3* with the WT (Table S6). Some targets were repressed in the WT in interaction with *A. alternata* (Fig. 12) and *R. solani* AG5 (Fig. S12), and their repression was lost in the *Δdcr2* and *Δago3* mutants. Moreover, in the *Δdcr2* mutant, we observed the overexpression of these predicted targets at the DC stage in the *Δdcr2* vs *A. alternata* interaction (Fig. 11; Fig. S11).

## Transport and allorecognition genes from *A. alternata* are potential targets of *T. atroviride*-siRNAs

To explore the potential cross-species regulation of *T. atroviride* small RNAs in the host fungus, we searched for target genes in the *A. alternata* genome. The analysis of target genes for the siRNAs induced in the WT strain at the DC and AC stages (Data set 2.12) uncovered 2,132 potential targets in *A. alternata* (Data set 2.12). A gene ontology enrichment analysis indicated that these targets were enriched in transport and localization, signaling and response to stimulus, fatty acid metabolism, and DNA replication (Data set 2.13). Within these targets, we also found 43 HET-domain-containing genes. In addition, we searched the *A. alternata* genome for targets of milR15, which was up-regulated at the AC stage in the WT strain. In this case, we found targets related to redox, transport, *het* genes, and genes encoding MFS transporters, glycoside hydrolases, and glycosyltransferases (Data set 2.14).

## DISCUSSION

A crucial component of *Trichoderma*'s mycoparasitism is its ability to recognize its host. Previous studies showed the induction of secondary metabolites, protein effectors, and lytic enzymes during mycoparasitism, some even before physical contact with the host (7, 12, 36). Our results indicate that *T. atroviride* distinguishes one host from another,

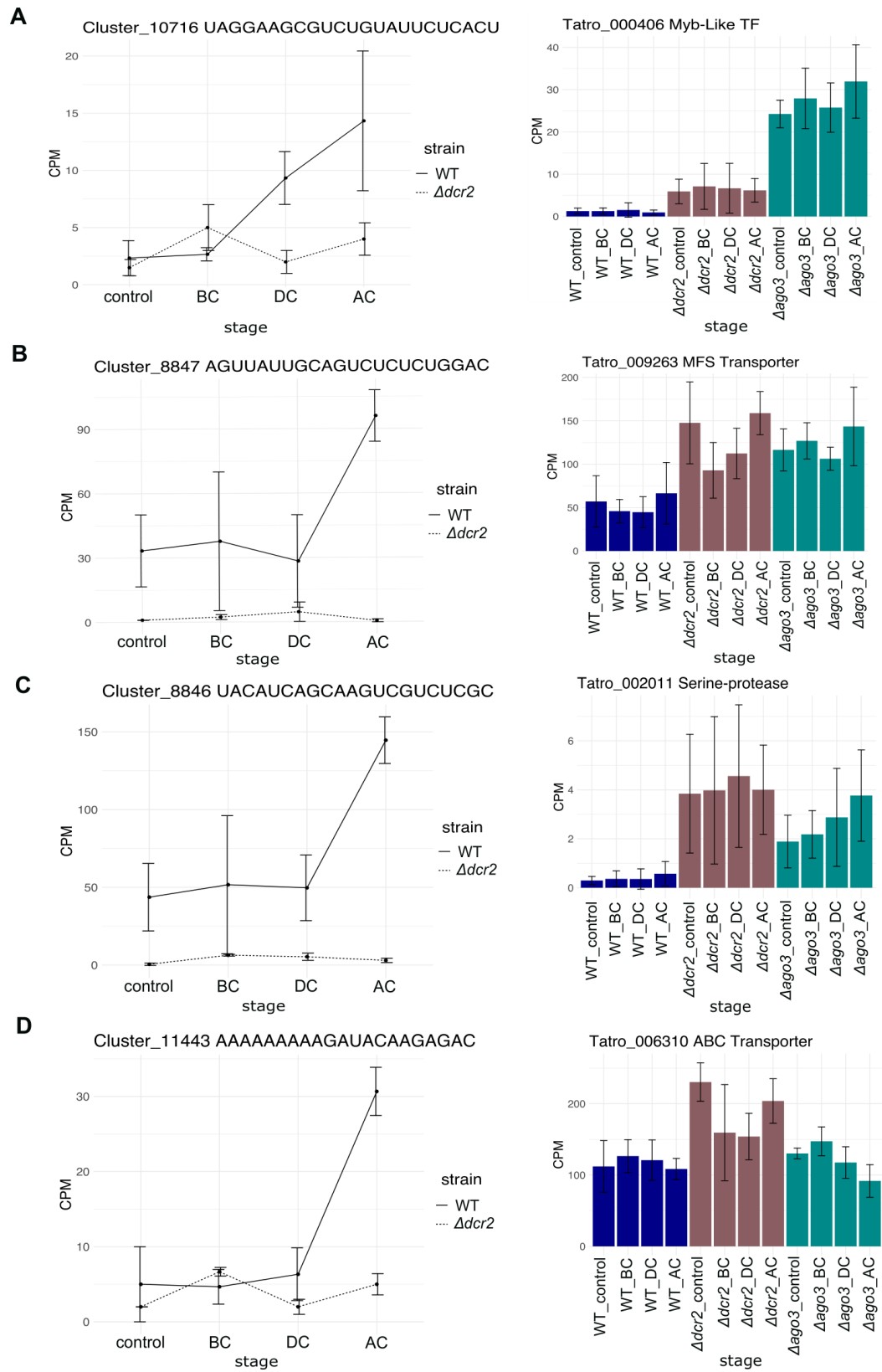

**FIG 11** Expression of siRNAs and their targets. Counts per million (CPM) of some up-regulated siRNAs (left plot) and the expression of their predicted targets (right plot) in RNA-seq libraries of WT (blue bars), Δ*dcr2* (red bars), and Δ*ago3* (green bars) strains.

## *Alternaria alternata*

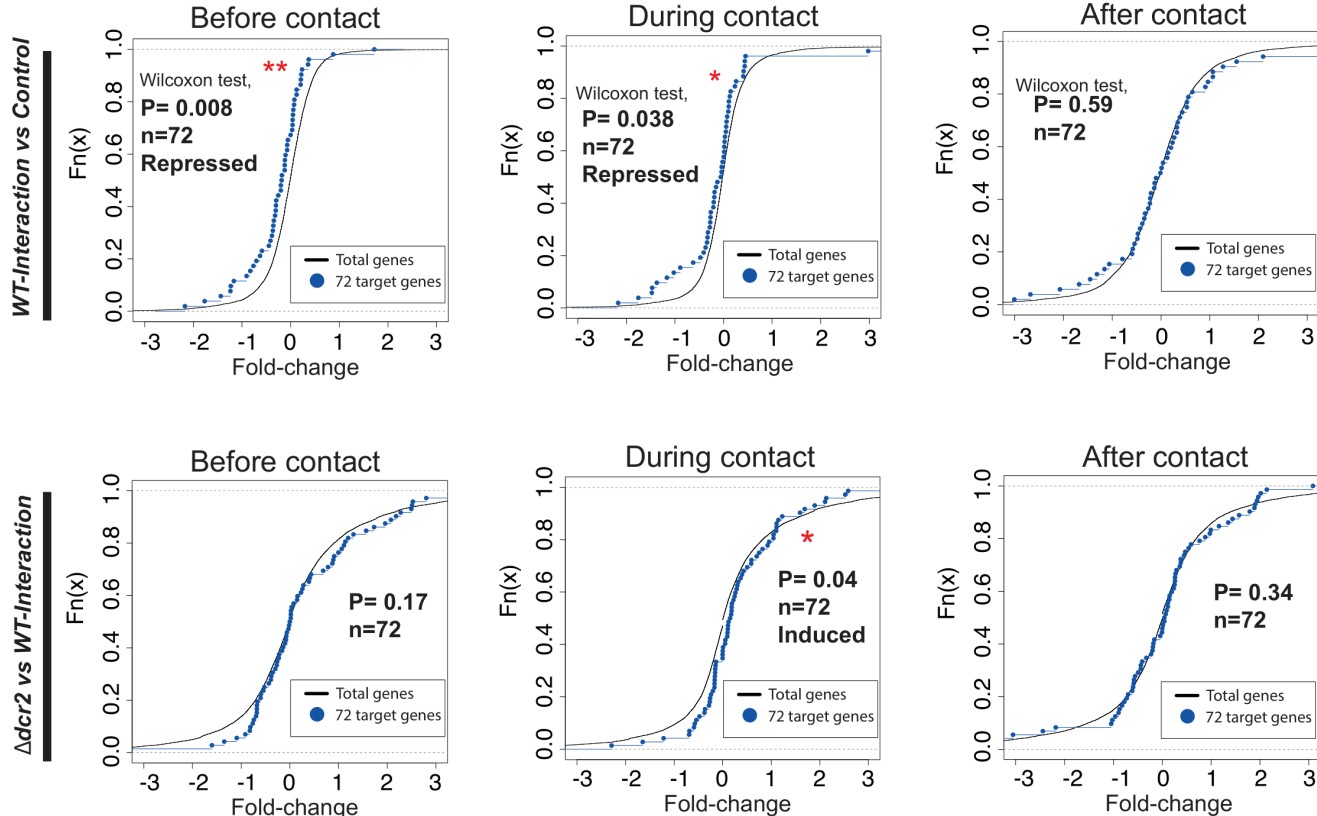

**FIG 12** Plots of predicted target genes against the total genes in the transcriptomic libraries. The upper panel shows the repression of the predicted targets in the WT vs *A. alternata* interaction. The lower panel shows the induction of the predicted targets in the *Δdcr2* vs *A. alternata* interaction.

likely through the perception of different signals. In our RNA-seq analysis, we found induced previously reported effectors (33, 37), highlighting a cerato-platanin, which is specific to the *T. atroviride-R. solani* AG5 interaction (stages DC and AC) and a CFEM domain gene induced in the *T. atroviride-A. alternata* interaction (BC stage).

Despite its ability to display a host-specific response, part of the *T. atroviride* transcriptional response is common to all host fungi. This response begins with chemical warfare characterized by early induction of secondary metabolism genes (PKSs and NRPS) and oxidation-reduction processes. During the overgrowth stage, the response shifts toward the induction of CAZymes, lipases, transport genes, and transcription factors. When interacting with *A. alternata, T. atroviride* employs a complex transcriptional strategy that involves a series of modular processes for survival against the host's diffusible metabolites, including ABC-type efflux pumps, enzymes for degradation, and recognition of foreign genetic material. This profile describes an actual "battle," and much of this response appears to depend on regulation by siRNAs.

The transcriptional response of the *Δdcr2* mutant during mycoparasitism sheds light on the role of Dcr2 in cell communication and reveals its importance in regulating protein, carbohydrate, and transport metabolism. Molecular transport is critical in fungi (38, 39), and its induction during mycoparasitism is highly relevant in *Trichoderma* species (40) and other mycoparasites like *Clonostachys rosea* (41). We found multiple up- and down-regulated transporters in this mutant, mainly of the MFS (122 genes) and ABC (19 genes) type, with one of the most affected network modules related to transport. ABC transporters are induced in *T. asperellum* in the presence of *A. alternata* toxins (42) and MFS-type transporters in *T. harzianum* in the presence of *B. cinerea*

cell walls (40). In the *Δdcr2* strain, the ABC multidrug resistance MDR transporters are repressed (11 genes) in more significant quantities than those induced (eight genes). ABC transporters are crucial in providing resistance to toxins. These results strongly suggest that Dcr2 participates in the toxin excretion process to survive the presence of antifungal compounds.

Moreover, some MFS-type transporters play a significant role in transporting sugars. For instance, the *T. atroviride* ortholog of RCO-3, an MFS involved in glucose transport and conidiation in *N. crassa* (43), repressed in the *Δdcr2* mutant when confronted with *A. alternata* is part of a glucose transport regulation and gene regulation module, which is required during the late stages of mycoparasitism. In these late stages, *T. atroviride* uptakes most nutrients from the host fungus that has already been killed. This lack of activation also occurs with other transporters, indicating that Dcr2 regulates transport of simple carbohydrates. Dcr2 is placed in the spotlight of mycoparasitism since a mutant in the corresponding gene cannot detoxify secondary metabolites secreted by the host fungus and shows a reduced capacity to use nutrients. In this sense, it was recently reported that Dicer plays an essential role in the mycoparasitic process of *Clonostachys rosea* on *B. cinerea* and *F. graminearum* (27). Thus, the importance of RNAi in fungus-fungus interactions extends beyond the genus *Trichoderma*.

Argonaute-3 is a positive regulator of amino acid biosynthesis, including leucine, lysine, and aromatic amino acids (Fig. S8). In this regard, *Trichoderma* species induce genes related to amino acid metabolism during the early stages of mycoparasitism (13, 44, 45). Amino acids are essential precursors for various secondary metabolites, such as non-ribosomal peptides like gliotoxin, peptaibols like alamethicin and tricho-zianinins, and pigments that play essential roles in biocontrol and, in general, in the capacity of fungi to protect themselves from adverse environments (45, 46). In addition, secondary metabolites are involved in interkingdom and intrakingdom encounters, and at physiologically relevant concentrations, they may act as signals rather than as toxins (41). Therefore, the observed repression of genes involved in amino acid metabolism in the *Δago3* mutant and that of the PKS and NRPS genes in *Δdcr2* represent a significant defect.

Effectors like hydrophobins are crucial in *Trichoderma's* interactions with plants and fungi (47, 48). In the *Δago3* strain, *hfbI* and *hfbII* were induced at the BC stage but repressed in the WT at this stage of the interaction and the DC stage. This suggests that Ago3 is also involved in the negative regulation of effectors. In addition, the gene co-expression network analysis revealed that the *Δago3* mutant is impacted in the blue module related to signaling, communication, and stress response. The hub gene of this module, the β-glucosidase SUN4 orthologue, decreases in expression in both mutants after contact, whereas its expression remains unchanged in the WT. This gene participates in cell wall remodeling and septation in other fungi (49).

The gene co-expression analysis showed that the expression of genes encoding cell wall-degrading enzymes is negatively affected in the *Δdcr2* and *Δago3* mutants in response to *A. alternata*. Previous studies in *T. reesei* have shown that some CWDEs, including cellulases and hemicellulases, may be regulated by milRNAs (50). *T. atroviride* may exhibit similar regulation mechanisms to produce cellulases and chitinases, essential for the parasitism of fungi and oomycetes.

Although Dcr2 appears to regulate most transport processes, other processes are regulated by both *dcr2* and *ago3*, such as specific MFS transporters and ROS response genes, including superoxide dismutase, which were repressed in both mutants at all stages. These findings suggest that Dcr2 and Ago3 co-regulate important genes involved in mycoparasitism.

Both prediction and Wilcoxon tests strongly suggest that siRNAs regulate RNAi targets, which is supported by the fact that they remain repressed in the WT strain, whereas in the *Δdcr2* mutant siRNAs disappear and their targets are overexpressed. The fact that RNAi targets are repressed in the WT strain when exposed to *R. solani* AG5 and *A. alternata* suggests a vital role in their regulation *via* siRNAs during mycoparasitism.

Notably, Myb-type transcription factors, transporters (MFS and ABC), and genes related to the regulation of proteolysis are among the targets. The importance of proper induction of ABC and MFS genes during mycoparasitism is evident, as they play essential roles in the process. Thus, uncontrolled induction, as observed in the Δdcr2, may be detrimental.

Because of the problems typically found in predicting small RNAs in fungi, such as the high number of false positives and a lack of consensus in prediction and annotation pipelines (51), further experimental validation is necessary to corroborate the function of the sRNAs found in this work. For milRNA prediction, we followed the criteria proposed by Johnson et al. (51), such as milRNAs abundance (cutoff 0.5 rpm) and strandedness (at least 80% of the aligned reads stranded), besides we performed three independent sequencing replicates. In the case of siRNAs, however, there are still no standard rules for their prediction and annotation in fungi. Nevertheless, although under other conditions, Tatro-milR-2 and Tatro-milR21 have already been experimentally validated in *T. atroviride* (34).

An interesting result is the possibility of the transfer of small RNAs from *T. atroviride* to the host fungi during mycoparasitism. In this regard, the presence of siRNAs and milR16 targets in the *A. alternata* genome indicates that they may impact the fitness of the host fungus silencing transporter and *het* genes. These results suggest that *T. atroviride* may silence transport and recognition in *A. alternata*, hindering its ability to respond effectively to the mycoparasitic threat, likely increasing *Trichoderma*'s ability to parasitize *A. alternata*. However, Tatro-milR16 and other small RNAs and their targets need to be experimentally validated in mycoparasitic conditions, and further work is essential to validate the transfer of small RNAs during mycoparasitism. The exploration of this potential phenomenon is not only exciting but also holds significant relevance in the field.

In summary, *T. atroviride* employs specific processes to respond to each phytopathogen while also needing to survive the toxic compounds released by the host fungi. Dcr2 is essential for prioritizing CAZymes activity, efflux transport, and regulation processes over primary metabolism. The interaction with *A. alternata* depends on small RNAs, with Ago3-carried small RNAs regulating amino acid metabolism and host recognition. Finally, siRNAs play a more important role in *T. atroviride* during mycoparasitism than milRNAs.

## MATERIALS AND METHODS

### Fungal strains and culture conditions

The IMI 206040 (WT) strain of *Trichoderma atroviride* and its respective mutant strains of the RNAi machinery (21, 34) were used. The phytopathogenic fungi used were *Alternaria alternata*, *Botrytis cinerea*, *Rhizoctonia solani* AG2 and AG5, *Fusarium spp.*, and *Fusarium oxysporum*. All the strains used in this work were cultured on potato dextrose agar (PDA, Difco; NJ, USA) at 27°C in the dark.

### Interaction in dual cultures

To evaluate the direct attack of *T. atroviride* against phytopathogenic fungi, we established dual cultures in 9 cm diameter Petri dishes with 25 mL of PDA. The strains were inoculated at one of the ends of the Petri dishes with a 0.5 cm mycelial plug of the *T. atroviride* strains. On the opposite side, 5 cm away, the plates were inoculated with a fragment of the host fungus' mycelium. Due to the different growth rates of the host fungi and some of the *T. atroviride* strains, they were inoculated at different times to balance colony diameters between the different fungal species, as follows: Inoculated plates with *A. alternata*, 6 days after its incubation, the *Trichoderma* strains were inoculated. For *Fusarium* spp. and *F. oxysporum*, the plates were inoculated and incubated for two days before inoculating the *Trichoderma* strains. For *R. solani* AG2 and AG5, the *Trichoderma* strains were inoculated first. After 12 hours of incubation,

the *R. solani* strains were inoculated, respectively. The *B. cinerea* strain was inoculated and incubated 26 hours before inoculation with the *Trichoderma* strains. The Δ*dcr2* and Δ*dcr1*Δ*dcr2* mutants were inoculated 24 hours before the mentioned times of the *Trichoderma* strains. At the end of the incubation time of each of the interactions, their growth was evaluated and photographically documented (Fig. S1). As controls, the WT and mutant strains of the RNAi machinery of *T. atroviride* and the phytopathogenic fungi without interaction were inoculated and incubated in individual Petri dishes. Assays were performed in triplicate.

## Antibiosis assays by soluble compounds

Antibiosis tests were conducted in 9 cm diameter Petri dishes with approximately 25 mL of PDA. The culture medium was covered with a 10 cm diameter layer of sterile cellophane and then inoculated with a 0.5 cm diameter fragment of mycelium from the *T. atroviride* strains in the center of the plate. The plates were incubated for 36 hours in the dark at 27°C. Then, the *T. atroviride* colony was removed by removing the cellophane layer, and immediately after, those same plates were inoculated in the center with 0.5 cm diameter fragments of mycelium of the phytopathogenic fungi. The plates were incubated for 36 hours in the dark at 27°C. The same methodology described above was followed to evaluate the *R. solani* AG5 and *A. alternata* diffusible compounds on the RNAi mutant strains, but first inoculating the phytopathogens.

## Messenger RNA-seq and small RNA-seq experiments during mycoparasitism

For the dual culture assays evaluating the different stages of mycoparasitism, the *T. atroviride*-WT strains and the Δ*dcr2* and Δ*ago3* mutants were inoculated in Petri dishes with PDA covered with a sterile cellophane sheet with a 0.5 cm fragment of mycelium, the fungal host strains *A. alternata*, *R. solani* AG2, and AG5 were inoculated at 5 cm distance. The mycelium of the *Trichoderma* strains was collected before contact (BC), at contact (C), and after contact (AC) during the confrontation with the fungal host, as shown in Fig. 1B. Stage BC was considered when the *T. atroviride* mycelium was 5 mm away from the host fungus, and stage AC was when *T. atroviride* reached an overgrowth of 5 mm on the fungal host. Mycelium was harvested under a red security light, frozen in liquid nitrogen, and stored until RNA extraction. The mycelium of the WT and mutant strains were collected simultaneously (40 hours after inoculation) as a control. Assays were performed three different times with three plates per replica.

## RNA isolation, library preparation, and sequencing

RNA was extracted using the TRIzol method, and RNA quality was assessed using the 2100 Bioanalyzer (Agilent). Only samples with a RIN value ≥8 were used. Total RNA was sent to the UGA-LANGEBIO Advanced Genomics Unit, Guanajuato, Mexico, for library preparation and sequencing. RNA libraries for sequencing were prepared according to Illumina TruSeq V2 RNA sample preparation (1 × 100) for RNA-Seq libraries and TruSeq small RNA libraries (1 × 36). 90 RNA-Seq libraries and 24 small RNA-Seq libraries were sequenced in the NovaSeq platform (Illumina). An average of 9 million raw reads per library was obtained for RNA-Seq, and 24 million raw reads per library were obtained for small RNA-Seq.

## RNA-Seq and small RNA-Seq data analysis

The quality of the RNA-seq data was evaluated using FastQC version 0.11.8 (52) and Trimmomatic (53) with default parameters to eliminate adapters and low-quality readings. Raw RNA-Seq data are available under Gene Expression Omnibus (GEO) accession number GSE190033. High-quality reads were aligned to the new *T. atroviride* genome (Atriztán-Hernández et al. in preparation) using HISAT2 (54). The reads were quantified using the "featureCounts" function of the Rsubread package (55). Genes with fewer than 10 reads were filtered out, and library size normalization was applied.

Differential expression analysis was done using the R packages Limma version 3.14 (56) and edgeR version 3.14 (57). Each library was compared to its control. DEGs were those with an FDR value ≤0.05. The topGO package version 2.46 (58) was used to perform enrichment analyses of selected gene pools ($P < 0.05$).

The codes used for RNA-Seq data analysis are available at https://github.com/elienriquez/RNA-Seq_analysis.

Small RNA-Seq libraries were processed using Cutadapt (59) to remove adapters and poor reads. Raw small RNA-Seq data are available under Gene Expression Omnibus (GEO) accession number GSE197448. High-quality reads were aligned to the new *T. atroviride* reference genome (Atriztán-Hernández et al., in preparation) using Bowtie version 1.2.3 (60). The reads mapped to rRNA, tRNA, snoRNA, snRNA, and lncRNA sequences (61) were removed. ShortStack software version 3.8.5 (62) was used to annotate, quantify, and predict milRNAs (parameters used: --dicermin 20 --dicermax 26 --mismatches 1 --foldsize 600). miRDeep2 was also used for the prediction of milRNAs (63). We only chose milRNAs that were present in at least two libraries. Differential expression analysis of small RNA clusters was done using the DESeq2 (64) and Rsubread package (55). The DE-siRNAs were those with FDR < 0.05 and logFC ± 1. The volcano plots of DE-siRNAs were performed using the EnhancedVolcano R package (65). Target gene prediction of identified sRNAs was carried out using TAPIR (66) and TargetFinder (67), and psRNATarget (68). The *A. alternata* reference genome was used to search for targets (69). The web service BlastKOALA was used to annotate the DEGs and predicted targets for the KEGG database (70).

The codes used for small RNA-Seq data analysis are available at https://github.com/elienriquez/smallRNA_seq_analysis.

## Construction of the gene co-expression network

We built a co-expression network using the WGCNA package (71). 90 RNA-Seq libraries of the interactions of the WT strain and the mutant strains of the RNAi machinery were used. We use a Pearson correlation model to build the adjacency matrix. The weighted parameter b was determined by the law of scale-free networks. A value b = 10 was used, the lowest value for which the scale-free topology index curve remains stationary when reaching a value of 0.9 (scale-free R2 = 0.9).

Furthermore, the weighted adjacency matrix was transformed into a topological overlap measure (TOM) matrix to estimate the connectivity in the network. The weighted network was of type "signed," which was built using the function "automatic mode" with the option of separate modules "unmerged." The genetic module size was set to 30 to obtain appropriate modules, and the threshold to merge similar modules was set to 0.25. Average linkage hierarchical clustering was used to construct a clustering dendrogram of the TOM matrix. The network generated by WGCNA was visualized using the Cytoscape v.3.9.1 program (72) with a threshold value of 0.9. Network metrics were obtained by Cytoscape. To obtain the biological functions of each of the modules, we performed an enrichment analysis in terms of GO in biological processes (BP) and molecular function (MF) using topGO version 2.46 (58). Hub genes were obtained as follows: Each gene with the highest intramodular connectivity in each network module was calculated by the WGCNA algorithm using the "chooseTopHubInEachModule" function, which was identified as a hub gene. Subsequently, the DEGs of each of the conditions were assigned to the co-expression network using Cytoscape v.3.9.1 (72) to compare the patterns of the modules in terms of their expression between the different conditions and stages of mycoparasitism.

## Biosynthetic cluster identification and domain prediction

The *T. atroviride* genome and its annotation files were retrieved from the Genbank (accession number: GCF_000171015.1) and analyzed with antismash v7 (73) to annotate regions rich in biosynthetic gene clusters for secondary metabolites. Then, we

identified core biosynthetic genes such as polyketide synthases and non-ribosomal peptide synthetases for each region. We then analyzed the protein domain composition of Tatro_011905 using the InterPro database (35). This analysis revealed the presence of six modules for amino acid incorporation, including an N-terminal A domain and a C-terminal cyclization domain. Given the number of modules and domain composition of this NRPS, we predicted that its product could be a cyclic hexadepsipeptide, as this domain organization resembles that of Destruxin, Isaridin, and Isariin (31) produced by *Beauveria felina*, a member of the Hypocreales family. Therefore, the protein sequence was then used as a query for a blast search against proteins from *B. felina*, known to produce cyclohexadepsipeptides (74). Among their hits, the NRPSs for the hexadepsi-peptides Destruxin, isaridin, and isariin were found with 40%, 36%, and 31% sequence homology.

## ACKNOWLEDGMENTS

We wish to thank Pedro Martínez-Hernández for his technical assistance.

This work was supported by Consejo Nacional de Ciencia y Tecnología (CONA-CyT) de México, grant numbers 929171 and 957501 to EEEF and CPS, respectively. This material by m-CAFEs Microbial Community Analysis & Functional Evaluation in Soils (m-CAFEs@lbl.gov), a Science Focus Area led by Lawrence Berkeley National Laboratory, is based upon work supported by the U.S. Department of Energy, Office of Science, and Office of Biological & Environmental Research under contract number DE-AC02-05CH11231. This work was supported by Grant FOINS-CONACYT (I0110/193/10FON.INST.-30–10) to A.H.-E.

## AUTHOR AFFILIATIONS

[1]Laboratorio Nacional de Genómica para la Biodiversidad-Unidad de Genómica Avanzada, Cinvestav Campus Guanajuato, Irapuato, Guanajuato, Mexico
[2]Novo Nordisk Center for Biosustainability, Technical University of Denmark, Lyngby, Denmark
[3]The LatAmBio Initiative, Irapuato, Guanajuato, Mexico
[4]Plant and Microbial Biology Department, University of California, Berkeley, Carlifornia, USA
[5]Environmental Genomics and Systems Biology Division, Lawrence Berkeley National Laboratory, Berkeley, Carlifornia, USA

## AUTHOR ORCIDs

José Manuel Villalobos-Escobedo  http://orcid.org/0000-0002-8412-6748
Alfredo Herrera-Estrella  http://orcid.org/0000-0002-4589-6870

## FUNDING

| Funder | Grant(s) | Author(s) |
| --- | --- | --- |
| Consejo Nacional de Ciencia y Tecnología (CONACYT) | 929171 | Eli Efrain Enriquez-Felix |
| Consejo Nacional de Ciencia y Tecnología (CONACYT) | 957501 | Camilo Pérez-Salazar |
| U.S. Department of Energy (DOE) | DE-AC02-05CH11231 | José Manuel Villalobos-Escobedo |
| Consejo Nacional de Ciencia y Tecnología (CONACYT) | I0110/193/10FON.INST.-30-10 | Alfredo Herrera-Estrella |

## AUTHOR CONTRIBUTIONS

Eli Efrain Enriquez-Felix, Conceptualization, Data curation, Formal analysis, Funding acquisition, Investigation, Project administration, Resources, Supervision, Validation,

Writing – original draft, Writing – review and editing | Camilo Pérez-Salazar, Data curation, Formal analysis, Investigation, Methodology, Validation, Writing – original draft, Writing – review and editing | José Guillermo Rico-Ruiz, Formal analysis, Investigation, Methodology, Writing – original draft, Writing – review and editing | Ana Calheiros de Carvalho, Formal analysis, Investigation, Writing – review and editing | Pablo Cruz-Morales, Formal analysis, Investigation, Methodology, Writing – review and editing | José Manuel Villalobos-Escobedo, Conceptualization, Formal analysis, Investigation, Methodology, Supervision, Writing – original draft, Writing – review and editing | Alfredo Herrera-Estrella, Conceptualization, Formal analysis, Funding acquisition, Investigation, Methodology, Project administration, Resources, Supervision, Writing – original draft, Writing – review and editing

## DATA AVAILABILITY

All RNA-seq data relevant to this publication are available from the NCBI's Gene Expression Omnibus database (GSE190033). All small RNA-seq data relevant to this publication are available from NCBI's Gene Expression Omnibus database (GEO: GSE197448).

## ADDITIONAL FILES

The following material is available online.

### Supplemental Material

**Data Set S1 (Spectrum03165-23-s0001.xlsx).** Lists of differentially expressed genes found in the comparisons between WT - fungal prey (Rhizoctonia solani AG2, R. solani AG5, and Alternaria alternata) versus WT strain growing alone at the different stages: before contact (BC), during contact (DC) and after contact (AC) 1-9. Differentially expressed genes found in the Δdcr2 and Δago3 strains in confrontation with the different fungal prey (Rhizoctonia solani AG2, R. solani AG5, and Alternaria alternata) compared to WT at the different stages (BC, DC, AC).
**Data Set S2 (Spectrum03165-23-s0002.xlsx).** Lists of genes and GO terms used to elaborate all main and supplemental figures.
**Supplemental figure and tables (Spectrum03165-23-s0003.pdf).** Twelve supplemental figures and seven supplemental tables.

### Open Peer Review

**PEER REVIEW HISTORY (review-history.pdf).** An accounting of the reviewer comments and feedback.

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
