## [Reviewer comments · Microbiology Spectrum]

Microbiology Spectrum

Argonaute and Dicer are essential for communication between *Trichoderma atroviride* and fungal hosts during mycoparasitism

Eli Enriquez-Felix, Camilo Pérez-Salazar, José Rico-Ruiz, Ana Calheiros de Carvalho, Pablo Cruz-Morales, José Villalobos-Escobedo, and Alfredo Herrera-Estrella

Corresponding Author(s): Alfredo Herrera-Estrella, Centro de Investigacion y de Estudios Avanzados del Instituto Politecnico Nacional

Review Timeline:

Submission Date:	August 23, 2023
Editorial Decision:	November 22, 2023
Revision Received:	January 10, 2024
Editorial Decision:	January 24, 2024
Revision Received:	February 2, 2024
Accepted:	February 17, 2024

Editor: Gustavo Goldman

Reviewer(s): Disclosure of reviewer identity is with reference to reviewer comments included in decision letter(s). The following individuals involved in review of your submission have agreed to reveal their identity: Nicole Donofrio (Reviewer #2)

Transaction Report:

DOI: <https://doi.org/10.1128/spectrum.03165-23>

Re: Spectrum03165-23 (Argonaute and Dicer are essential for communication between *Trichoderma atroviride* and fungal hosts during mycoparasitism)

Dear Dr. Alfredo H Herrera-Estrella:

Thank you for the privilege of reviewing your work. Below you will find my comments, instructions from the Spectrum editorial office, and the reviewer comments.

I apologize for the delay but it was difficult to get two reviewers to evaluate your manuscript. Both reviewers have provided several suggestions that can improve your manuscript. Please, submit a revised version together with a rebuttal letter addressing point-by-point raised by each reviewer.

Revision Guidelines

Sincerely,
Gustavo H. Goldman
Editor
Microbiology Spectrum

Reviewer #1 (Comments for the Author):

In this study the authors explore the role of the RNAi machinery of *T. atroviride* in control of mycoparasitism of a number of fungal host species. They show that particular RNAi deletion mutants (Dcr2 and Ago3) struggle to manage the interactions with

other fungal hosts, resulting in varied efficiency of mycoparasitism. The authors collect a number of interesting datasets, including multiple RNA-seq and small RNA-seq datasets and extensively analyze these datasets to gain insight into this regulation. They conclude that particular RNAi components are responsible for different aspects of the transcriptional regulation that occurs before, during, and after interaction of *T. atroviride* with fungal hosts.

Overall, the topic is important and the science is intriguing, but sometimes the story is lost due to an overwhelming amount of information. In places, the data presentation could be improved to aid readability. The discussion of the GO analyses is particularly dense, and might benefit from some additional structure, or removal of unessential details. Specific comments are listed below:

Major Comments

1. Some more information about the RNAi machinery of *T. atroviride* would be helpful in the introduction. How many orthologs exist in *T. atroviride*, are they all functional, etc?
2. Regarding Figure 1 (and S1). The labels are on the left side for the organisms streaked out on the right side of the plate. Could the labels be swapped or the images rotated to make it easier for the reader? The legend has an error on line 805-806, where it states that both organisms were plated on the left. Finally, the conclusions made in the text are not always clear from the images, for example regarding the changes observed with the *dcr* double knockout and *R. solani* AG2. Maybe a few more words describing the phenotype(s) would be useful? Alternatively, arrowheads could be added strategically to the figure to point out particular focal points. Panel B is a step in this direction.
3. Figure 1C, could the right side be labeled as having received diffusible compounds? Are there methods for these experiments listed somewhere? In general, the description of the experiments is often lacking, making assessment of the results challenging.
4. Along this line, it is not clear from the text or legend what is being shown in Figure 1D-E. A few more details would be helpful to the reader to help justify the conclusion on lines 131-132. For example, are the diffusible compounds known?
5. Figure 2A suffers from overplotting. The author should consider two-dimensional PCA plots or another plot type where the data can be more easily displayed.
6. Line 218-225 do not provide any real description of the data shown in Figure 6. Should we know anything else about these genes and their biology? If not, this section could be reduced/removed. Relatedly, Figure 7 does not really add anything to the main text, and might be better suited in the supplement.
7. Figure 9, again the figure suffers from overplotting. Consider removing the names and possibly adding more detail to the figure legend about the construction of the network. Details are often scattered across the methods, legends, and text, making it an effort for the reader to find the relevant information.
8. The final section of the results hints at the idea of interorganismal transfer of small RNAs. This is a challenging topic/field that is plagued by poor demonstrations of RNA transfer. For this to be a valuable contribution to the field, the authors should either prove that the small RNAs are transferred from one organism to the other, or consider removing this results section. Predictions alone do not demonstrate functional transfer of RNA between organisms. Perhaps this would be better suited as a discussion point?
9. In the discussion, the authors should explain the challenges of the study, including where error or misinterpretation might be present. As just one example, small RNA target prediction results in a high rate of false positives. How could this influence the findings of this study?

Minor Comments

10. Line 131-132, the authors write several times that *Dcr2* detoxifies diffusible metabolites, but it seems more likely that *Dcr2* regulates detoxification pathways. More careful wording could be applied.
11. Line 114, it would be helpful to mention that the wild type is *T. atroviride* at the start of the intro.
12. Figure 3, what are the colors on the left side of the heatmap (other heat maps as well)? Also, the legend mentions right and left panels, but the figure shows A and B.
13. Line 308-9, have the siRNAs been previously defined? If not, how were these small RNAs classified as siRNAs? Did they try to validate that any of these small RNAs are expressed, e.g., by Northern blot, or confirm any of the potential targets?
14. Line 339, should this refer to Figure S8 as stated or S7?
15. Line 399-400 seems to lack a reference.
16. Line 815, white and black bars should be updated to correct colors.
17. Figure 3A, LogFC in the figure should be on one line.

Reviewer #2 (Comments for the Author):

In their manuscript entitled *Argonaute and Dicer are essential for communication between Trichoderma atroviride and fungal hosts during mycoparasitism*, the authors do a tremendous amount of work to decipher gene expression and regulation before contact, during contact and after contact in dual-organism plate assays. They carefully analyze their data to confirm the role of *Dicer2* and *argonaute-3* (small RNA machinery) in these stages, which adds new knowledge to the mechanism of biocontrol in *Trichoderma*. With the exception of some organizational and background aspects (marked with asterisks below), this represents an expansive amount of work, but with some very intriguing highlights (i.e. the current Figure 7), that add to how this important fungus does its job.

Importance

Line 50: consider starting the sentence with "There is an increasing need for plant disease control..."

In this paragraph, consider a word other than "use", which is found multiple times in a couple sentences.

Introduction

Line 74: is "a prey" the right term here? Maybe "before contact with a host fungus", or "before contact with their prey"

Line 95: I would include *Magnaporthe oryzae* in here, too (and related references). Also in this section, I would at least introduce the general roles of Dicer and Argonaute, given their gene abbreviations are mentioned in the next section.

Line 97: I think I am looking for a little more of a segue into WHY you wanted to look at RNAi machinery... what was the leading hypothesis?

Results

* for the results section, I suggest that at the beginning, mention where the *T. atroviride* RNAi strains came from - they are referenced but that is it. Even a table of the strains (all of them, including the plant pathogens), and their origins, would be helpful and would alleviate some of the confusion I am getting when I am looking at the figures. In this table or text, you could indicate that the fungal plant pathogens are "prey" and the *T. atroviride* WT and mutant strains are "predator" (although I am not sure those terms totally apply here, you could use something like those terms to differentiate them).

* I think overall, figures 1C-D are a little confusing as to what you are trying to show. Please make sure the text in the results section matches the correct figure and that the corresponding info in the legends are correct. Also, why did you test the *Rhizoc* diffusibles but not the other fungi? Especially since you mention early on that *T. atroviride* struggled against *A. Alternaria*. Would that not indicate diffusibles probably coming from *A. alternaria*, too?

Line 117: Instead of "WT strain", please say "the *T. atroviride* WT strain", otherwise, it seems like "WT" refers to the AG2 and Ag5 fungi.

Line 177: change "it was harder to colonize" to something like "it struggled against *A. alternata*" or "it took longer to grow over *A. alternata*". The way it's currently phrased is awkward.

Line 118: dcr2 and double dicer mutant of what? *T. atroviride*? Please specify, in order for the reader to keep the fungal names clear.

Figure S1 needs to be clarified, similar to the figure legend for Figure 1A... what is the top row?

Line 122: keep the mutant names consistent... if you use a delta symbol in front of some of them, please do it with all of them. Here, the ago mutants have no delta symbols.

Line 127: Here, I would mention briefly how you generated Figure 1C and S2. Perhaps just stating "a cellophane assay", or briefly stating what you describe in the MM section, is fine. Finally, do you mean Figure 1D here? I think 1C is the *Rhizoc* diffusibles, right?

Line 131-132: This statement seems a little bit over-reaching to me... I think this goes back to my earlier comment that in the introduction, it would be good to provide a bit of background on Dicer2 and its importance. I know it cannot all be covered, but just some general function statements. I think when you mess with Dicer2, there are a lot of indirect effects, and this could be one of them.

Line 138: here, I really like that you use the term "host" to define the fungal plant pathogens. Maybe use "host" instead of prey, throughout.

Perhaps for figure 4, you could incorporate the "vs." (for versus) somewhere into that figure at the bottom? It almost reads like you are looking at ago and dicer mutants of *Rhizoc* and *Alternaria*, instead of looking at gene expression in *T. atroviride* versus these plant pathogens. It somehow should be made more clear, so people can look at the figure and understand exactly what is happening.

Line 200: there is some italicized wording here that should not be italicized.

Line 206: please remind the reader what this phenotype was.

*Line 221-225, Figure 7: I understand that this is a very interesting finding. However, I am wondering if the figure, as it stands, needs to be in the main body. It might be better as a supplementary figure, since in my reading of the manuscript, protein structure is not really fitting in with all of the gene expression analyses and plate assays that come before. It seems very interesting, yet disjointed at the same time. One thought is this - could you make Figure 6 supplementary, but pulling out the one graph for this particular NRPS, and then combine that graph with Figure 7?

Line 231: "*A. alternaria*" should be italicized.

Line 234: this sentence should have references, and more specifics... are these known effectors in *Trichoderma*? Other fungi?

Line 244: change to "positively regulates"

Line 280-282: this sentence, as written, is unclear. Consider breaking it up into two sentences for clarity.

Line 304: Are these from the BC, DC or AC conditions?

Lines 337-345: I really enjoy and appreciate this part, using the data from the siRNA libraries to sort of confirm expression levels from the transcriptome data. It nicely confirms that the small RNA machinery is really at play here (and sort of negates my earlier comment about the Dicer2 statement being an "over-reach"!

Dear Editor,

We thank the reviewers for their constructive comments, which have certainly helped improve our manuscript. Below, you will find a point-by-point rebuttal to the reviewer's comments.

Responses to Reviewer #1:

Major comments

Q1. Some more information about the RNAi machinery of *T. atroviride* would be helpful in the introduction. How many orthologs exist in *T. atroviride*, are they all functional, etc?

R1. We agree with the reviewer that more information on the RNAi machinery of *Trichoderma atroviride* was necessary. Accordingly, we have added a couple of sentences in the introduction as follows:

Furthermore, the relevance of small RNAs in mycoparasitism has not been studied so far in T. atroviride. The T. atroviride genome encodes two Dicer, three Argonaute, and three RNA-dependent RNA polymerases. Previously, we reported that Dcr2 (Dicer) and Rdr3 are involved in the regulation of asexual development and that Dcr2 processes most of the sRNAs of 21-22 nucleotides, which are the most abundant sRNAs in T. atroviride (21).

Q2. Regarding Figure 1 (and S1). The labels are on the left side for the organisms streaked out on the right side of the plate. Could the labels be swapped or the images rotated to make it easier for the reader? The legend has an error on line 805-806, where it states that both organisms were plated on the left. Finally, the conclusions made in the text are not always clear from the images, for example regarding the changes observed with the dcr double knockout and *R. solani* AG2. Maybe a few more words describing the phenotype(s) would be useful? Alternatively, arrowheads could be added strategically to the figure to point out particular focal points. Panel B is a step in this direction.

R2. We agree with the reviewer that our former Figure 1 was not as clear as we wished. We have modified the figure and the corresponding legend. Briefly, we re-organized the figure and highlighted the areas we wanted the reader to focus on using dashed rectangles (see new Figure 1).

Q3. Figure 1C, could the right side be labeled as having received diffusible compounds? Are there methods for these experiments listed somewhere? In general, the description of the experiments is often lacking, making assessment of the results challenging.

R3. As mentioned in the previous response, we reorganized the figure for clarity. In the new Figure 1, the panel in question is now panel (D), which now includes the words control and treatment to differentiate them. In addition, we have included a better description of the experiments in the results section, which now reads:

Due to the significance of antibiosis in biocontrol mediated by Trichoderma, we evaluated the impact of diffusible compounds released by the RNAi mutant strains on the growth of various fungal species (Figs. 1C and S2). Briefly, we grew the different T. atroviride strains on PDA plates covered by a cellophane sheet to allow the removal of Trichoderma and diffusion of metabolites and then inoculated the indicated fungal hosts. All mutant strains inhibited the growth of the different fungal hosts to a similar degree to that observed for the WT, except for $\Delta rdr2$, which showed a strongly reduced capacity to inhibit the growth of all tested hosts (Fig. S2). Conversely, the effect of diffusible compounds from a fungal host (R. solani AG5 and A. alternata) on the growth of the RNAi mutant strains was evaluated (Figs. 1D & E and S3 & 4). Interestingly, the $\Delta dcr2$ and the $\Delta dcr1\Delta dcr2$ double mutant showed a significant growth reduction compared to the WT (Figs. 1D-E). These results suggest that the Dcr2 component regulates genes involved in detoxifying diffusible metabolites.

We also improved the description of the methods in the section “**Interaction in dual cultures**”.

Q4. Along this line, it is not clear from the text or legend what is being shown in Figure 1D-E. A few more details would be helpful to the reader to help justify the conclusion on lines 131-132. For example, are the diffusible compounds known?

R4. We agree with the reviewer that our description our somewhat poor. Therefore, we included a few lines in the results section, which now read:

All mutant strains inhibited the growth of the different fungal hosts to a similar degree to that observed for the WT, except for $\Delta rdr2$, which showed a strongly reduced capacity to inhibit the growth of all tested hosts (Fig. S2). Conversely, the effect of diffusible compounds from a fungal host (R. solani AG5 and A. alternata) on the growth of the RNAi mutant strains was evaluated (Figs. 1D & E and S3 & 4). Interestingly, the $\Delta dcr2$ and the $\Delta dcr1\Delta dcr2$ double mutant showed a significant growth reduction compared to the WT (Figs. 1D-E). These results suggest that the Dcr2 component regulates genes involved in detoxifying diffusible metabolites.

Q5. Figure 2A suffers from overplotting. The author should consider two-dimensional PCA plots or another plot type where the data can be more easily displayed.

R5. The figure was modified in a way that we consider displays the data in a friendlier way.

Q6. Line 218-225 do not provide any real description of the data shown in Figure 6. Should we know anything else about these genes and their biology? If not, this section could be reduced/removed. Relatedly, Figure 7 does not really add anything to the main text, and might be better suited in the supplement.

R6. We strongly believe Figures 6 & 7 are very relevant for our manuscript. However, as indicated by the reviewer, we had not described them well enough. Furthermore, in the opinion of reviewer#2, Figure 7 displays interesting findings. We have, therefore, reorganized Figures 6 & 7 and included an improved description of the findings in the results section, which now reads:

*Finally, due to the relevance of secondary metabolism in mycoparasitism (4, 14), we performed a prediction of biosynthetic gene clusters in *T. atroviride* to search for genes whose expression may be affected in the $\Delta dcr2$ strain. We identified core biosynthetic genes such as PKS and NRPS in 33 biosynthetic gene clusters (BGC). We observed evident defects in the expression of three PKSs and three NRPS belonging to three different BGCs in the $\Delta dcr2$ strain. These genes were expressed in the wild-type strain in all interactions but repressed in the $\Delta dcr2$ mutant (Fig. 6). Remarkably, within BGC 7.1, we found a gene (*Tatro_010797* gene) with a domain organization like that previously described in the NRPSs involved in the biosynthesis of the insecticidal hexa-depsipeptides destruxin, isaridin and isariin (Fig. 7A), all found in *Hypocreales* of the genus *Beauveria* (31). This gene is repressed in the $\Delta dcr2$ mutant in both mycoparasitic and control conditions (7B).*

Q7. Figure 9, again the figure suffers from overplotting. Consider removing the names and possibly adding more detail to the figure legend about the construction of the network. Details are often scattered across the methods, legends, and text, making it an effort for the reader to find the relevant information.

R7. We agree with the reviewer. The figure was modified as suggested, and the names were removed. We included a better description (see materials and methods; section **Construction of the gene co-expression network**) of how the network was obtained. The legend of the figure was also modified, and more details were added.

Figure 9. Weighted Gene Co-expression Network of the transcriptomic data from WT and RNAi mutant strains. (A) The figure shows the gene co-expression network constructed using

WGCNA with $\beta= 10$. Fifteen modules were identified with a total of 7326 genes. The colors of each node represent the module to which they belong. The modules with the highest number of genes are turquoise (1359 genes), blue (1194 genes), brown (1182 genes), yellow (560 genes), green (427 genes), and pink (362 genes). The network was visualized in the Cytoscape using a threshold value of 0.19. The nodes represent each gene, and the edges show the connection based on each gene's Correlation value (Pearson) according to its co-expression value. The most statistically significant ($P < 0.05$) GO terms for each module are shown. BP: Biological Process. MF: Molecular Function. The largest circle represents the hub gene of the module. (B) Heatmap of the different modules and their expression during the confrontation against the fungal hosts. The up- or down-regulation ($P < 0.05$) of each module in the $\Delta dcr2$ and $\Delta ago3$ strains is relative to the WT strain and is indicated by colors (red and blue, respectively).

Q8. The final section of the results hints at the idea of intra organismal transfer of small RNAs. This is a challenging topic/field that is plagued by poor demonstrations of RNA transfer. For this to be a valuable contribution to the field, the authors should either prove that the small RNAs are transferred from one organism to the other, or consider removing this results section. Predictions alone do not demonstrate functional transfer of RNA between organisms. Perhaps this would be better suited as a discussion point?

R8. We agree with the reviewer that the idea of inter-organismal transfer of sRNAs is challenging, and as such, it warrants further experimentation. However, we and reviewer#2 think our findings are interesting and deserve to be mentioned. Therefore, we did not remove this section but softened our conclusions only to suggest the possibility of small RNA transfer between *T. atroviride* and *A. alternata*. The description now reads:

Transport and allorecognition genes from A. alternata are potential targets of T. atroviride-siRNAs

To explore the potential cross-species regulation of T. atroviride small RNAs in the host fungus, we searched for target genes in the A. alternata genome. The analysis of target genes for the siRNAs induced in the WT strain at the DC and AC stages (Data set 2.12) uncovered 2,132 potential targets in A. alternata (Data set 2.12) revealed 2,132 possible targets. A gene ontology enrichment analysis indicated that these targets were enriched in transport and localization, signaling and response to stimulus, fatty acid metabolism, and DNA replication (Data set 2.13). Within these targets, we also found 43 HET-domain-containing genes. Additionally, we searched the A. alternata genome for targets of miR15, which was up-regulated at the AC stage in the WT strain. In this case, we found targets related to redox, transport, het genes, and genes encoding MFS transporters, glycoside hydrolases, and glycosyltransferases (Data set 2.14).

Q9. In the discussion, the authors should explain the challenges of the study, including where error or misinterpretation might be present. As just one example, small RNA target prediction results in a high rate of false positives. How could this influence the findings of this study?

R9. As suggested by the reviewer, the new version of the manuscript includes a paragraph explaining the challenges and limitations of our study, which reads:

*Because of the problems typically found in predicting small RNAs in fungi, such as the high number of false positives and a lack of consensus in prediction and annotation pipelines (50), further experimental validation is necessary to corroborate the function of the sRNAs found in this work. For miRNA prediction, we followed the criteria proposed by Johnson et al. (50), such as miRNAs abundance (cutoff 0.5 rpm) and strandedness (at least 80% of the aligned reads stranded), besides we performed three independent sequencing replicates. In the case of siRNAs, however, there are still no standard rules for their prediction and annotation in fungi. Nevertheless, although under other conditions, Tatro-miR-2 and Tatro-miR21 have already been experimentally validated in *T. atroviride* (34).*

Minor Comments

Q10. Line 131-132, the authors write several times that Dcr2 detoxifies diffusible metabolites, but it seems more likely that Dcr2 regulates detoxification pathways. More careful wording could be applied.

R10. We agree with the reviewer that the wording we used was misleading. Therefore, we modified the sentence to read:

These results suggest that the Dcr2 component regulates genes involved in detoxifying diffusible metabolites.

Q11. Line 114, it would be helpful to mention that the wild type is *T. atroviride* at the start of the intro.

R11. We mention in the introduction that we worked with mutants in RNAi machinery and the wild-type *Trichoderma atroviride*. The text reads:

*Screening with mutants in all components of the RNAi machinery and the WT strain of *T. atroviride* in confrontation with six phytopathogenic fungi indicated that this machinery plays a significant role in mycoparasitism.*

In addition, now the first sentence of the results section indicates that we worked with the *T. atroviride* WT and mutant strains as follows:

To evaluate the biocontrol capacity of the T. atroviride WT and RNAi mutant strains, we performed dual confrontation assays against Alternaria alternata, Fusarium oxysporum, Fusarium spp., B. cinerea, and two strains of R. solani, one belonging to anastomosis group 2 (AG2) and one to group 5 (AG5) (Table 1).

Q12. Figure 3, what are the colors on the left side of the heatmap (other heat maps as well)? Also, the legend mentions right and left panels, but the figure shows A and B.

R12. We are sorry for that mistake. We have modified the legend to figures 3 & 4 to read:

Figure 3. Differentially expressed genes and Gene Ontology functional categories from *T. atroviride* (WT) interactions. (A) The figure shows a heatmap of DEGs for all confrontations and stages BC, DC, and AC. Roman numerals represent the clusters formed by Pearson correlation. *The different clusters identified in the dendrogram are represented by the colored bar at the left of the heatmap.* (B) GO enrichment (Biological processes) of genes in clusters V, VIII, IX, X, XII, XIII, XIV, and XV (Right). The genes in these clusters are listed in Data Set 2.1.

Figure 4. Differentially expressed genes DEGs of RNAi mutant strains in confrontation with *R. solani* AG2/AG5 and *A. alternata*. (A) Heatmap showing the DEGs in $\Delta dcr2$ and $\Delta ago3$ mutant strains ($P < 0.05$) relative to the WT strain at three different stages of mycoparasitism (BC, DC, and AC). The black delimited rectangles are gene clusters shared between mutants. *The different clusters identified in the dendrogram are represented by the colored bar at the left of the heatmap.* The gene clustering was performed by correlation using the Pearson method. The GO terms found in the clusters are detailed in Data Set 2.5. (B) The graph shows the number of differentially expressed genes DEGs in $\Delta dcr2$ and $\Delta ago3$ compared to the WT strain. The up-regulated and down-regulated genes are represented by red and blue bars, respectively.

Q13. Line 308-9, have the siRNAs been previously defined? If not, how were these small RNAs classified as siRNAs? Did they try to validate that any of these small RNAs are expressed, e.g., by Northern blot, or confirm any of the potential targets?

R13. We modified the text to refer to them as small RNAs, not siRNAs.

The most considerable T. atroviride fraction of small RNAs is generated from intergenic regions. The size distribution of the small RNAs from the WT strain showed peaks of 21 and 22 nucleotides (Fig. 10A).

Q14. Line 339, should this refer to Figure S8 as stated or S7?

R14. It should have referred to Figure S7, which in the new version of the manuscript corresponds to Figure S11. The text reads now:

We found that some targets, including transcription factors, peptidases, and transporters, are indeed overexpressed in $\Delta dcr2$ (Figs. 11 and S11), indicating that the loss of the siRNA in $\Delta dcr2$ affects their expression.

Q15. Line 399-400 seems to lack a reference.

R15. Thank you for pointing this out; the sentence should have referred to a supplemental figure that we forgot to mention. This has been corrected, and the text reads:

*Argonaute-3 is a positive regulator of amino acid biosynthesis, including leucine, lysine, and aromatic amino acids (Fig. S8). In this regard, *Trichoderma* species induce genes related to amino acid metabolism during the early stages of mycoparasitism (13, 43, 44).*

Q16. Line 815, white and black bars should be updated to correct colors

R16. Thank you again for pointing out this mistake. We corrected the legend to Figure 1, which now reads:

*(E) Colony diameter of *T. atroviride* RNAi mutant strains growing in the presence of *R. solani* AG5 diffusible compounds (blue bars) or without them (red bars).*

Q17. Figure 3A, LogFC in the figure should be on one line.

R17. Once more, thank you for pointing out this mistake. The figure was modified accordingly.

Responses to Reviewer #2:

Q1. Line 50: consider starting the sentence with "There is an increasing need for plant disease control..." In this paragraph, consider a word other than "use", which is found multiple times in a couple sentences.

R1. Thank you for the suggestion. We have followed your advice and modified the text to read as follows:

There is an increasing need for plant disease control without chemical pesticides to avoid environmental pollution and resistance, and the health risks associated with the application of pesticides are increasing. Employing Trichoderma species in agriculture to control fungal diseases is an alternative plant protection strategy that overcomes these issues without utilizing chemical fungicides. Therefore, understanding the biocontrol mechanisms used by Trichoderma species to antagonize other fungi is critical. Although there has been extensive research about the mechanisms involved in the mycoparasitic capability of Trichoderma species, there are still unsolved questions related to how Trichoderma regulates recognition, attack, and defense mechanisms during interaction with a fungal host. In this work, we report that the argonaute and dicer components of the RNAi machinery and the small RNAs they process are essential for gene regulation during mycoparasitism by Trichoderma atroviride.

Q2. Introduction. Line 74: is "a prey" the right term here? Maybe "before contact with a host fungus", or "before contact with their prey"

R2. We replaced the word "prey" with "host fungus" throughout the manuscript.

Q3. Line 95: I would include *Magnaporthe oryzae* in here, too (and related references). Also in this section, I would at least introduce the general roles of Dicer and Argonaute, given their gene abbreviations are mentioned in the next section.

R3. Thank you for the suggestion. We now mention the *M. oryzae* research and added more details about RNAi core components. The paragraph now reads:

RNA interference (RNAi) is a highly conserved mechanism in eukaryotic organisms. This mechanism regulates gene expression through small RNAs (sRNAs) of approximately 20 to 30 nucleotides (15). sRNAs play an essential role in gene silencing at the transcriptional and post-transcriptional levels (16). The core components of RNAi are Dicer, an RNase III that processes long double-stranded RNA (dsRNA) into 21-26 nucleotide RNA fragments, and RNA-dependent RNA polymerases that convert aberrant RNA into dsRNA that Dicer proteins can process. The dsRNA fragments produced by Dicer are loaded in argonaute proteins, which form part of the RNA-induced silencing complex (RISC) (17). In Neurospora crassa, Fusarium graminearum, Mucor circinelloides, T. atroviride, Clonostachys rosea, Magnaporthe oryzae, Verticillium dahliae, Metarhizium robertsii, and Sordaria macrospora, mutants in the RNAi machinery present morphological and reproductive alterations, suggesting that these components regulate vegetative processes, hyphal development, and spore production (18-27). In other fungi, such as B. cinerea, R. solani, and Beauveria bassiana, the RNAi machinery is involved in processes

related to pathogenesis (28–30). Therefore, we wondered if small RNAs could exert part of this complex gene regulation. Furthermore, the relevance of small RNAs in mycoparasitism has not been studied so far in T. atroviride. The T. atroviride genome encodes two Dicer, three Argonaute, and three RNA-dependent RNA polymerases. Previously, we reported that Dcr2 (Dicer) and Rdr3 are involved in the regulation of asexual development and that Dcr2 processes most of the sRNAs of 21-22 nucleotides, which are the most abundant sRNAs in T. atroviride (21).

Q4. Line 97: I think I am looking for a little more of a segue into WHY you wanted to look at RNAi machinery... what was the leading hypothesis?

R4. As stated in the manuscript (lines x-Y), research in Trichoderma species indicates that transcriptional reprogramming plays an important role in mycoparasitism, which could involve posttranscriptional regulation and, therefore, the RNAi machinery. Nevertheless, such a possibility had not been explored. In addition, we included a few more lines on why we explored this possibility in a paragraph that reads:

In other fungi, such as B. cinerea, R. solani, and Beauveria bassiana, the RNAi machinery is involved in processes related to pathogenesis (28–30). Therefore, we wondered if small RNAs could exert part of this complex gene regulation. Furthermore, the relevance of small RNAs in mycoparasitism has not been studied so far in T. atroviride. The T. atroviride genome encodes two Dicer, three Argonaute, and three RNA-dependent RNA polymerases. Previously, we reported that Dcr2 (Dicer) and Rdr3 are involved in the regulation of asexual development and that Dcr2 processes most of the sRNAs of 21-22 nucleotides, which are the most abundant sRNAs in T. atroviride (21).

Q5. Results. * for the results section, I suggest that at the beginning, mention where the *T. atroviride* RNAi strains came from - they are referenced but that is it. Even a table of the strains (all of them, including the plant pathogens), and their origins, would be helpful and would alleviate some of the confusion I am getting when I am looking at the figures. In this table or text, you could indicate that the fungal plant pathogens are "prey" and the *T. atroviride* WT and mutant strains are "predator" (although I am not sure those terms totally apply here, you could use something like those terms to differentiate them).

R5. We added a table (Table 1) mentioning the origin of the strains used in our work. We also changed the word "prey" to "host fungus"

Q6. * I think overall, figures 1C-D are a little confusing as to what you are trying to show. Please make sure the text in the results section matches the correct figure and that the

corresponding info in the legends are correct. Also, why did you test the Rhizoc diffusibles but not the other fungi? Especially since you mention early on that *T. atroviride* struggled against *A. Alternaria*. Would that not indicate diffusibles probably coming from *A. alternaria*, too?

R6. We added a better description of this section. We also added a supplemental figure on the effect of diffusible compounds of *A. alternata* on the growth of RNAi mutant strains of *T. atroviride* (Fig S4).

*Due to the significance of antibiosis in biocontrol mediated by Trichoderma, we evaluated the impact of diffusible compounds released by the RNAi mutant strains on the growth of various fungal species (Figs. 1C and S2). Briefly, we grew the different T. atroviride strains on PDA plates covered by a cellophane sheet to allow the removal of Trichoderma and diffusion of metabolites, and then we inoculated the indicated fungal hosts. All mutant strains inhibited the growth of the different fungal hosts to a similar degree to that observed for the WT, except for $\Delta rdr2$, which showed a strongly reduced capacity to inhibit the growth of all tested hosts (Fig. S2). Conversely, the effect of diffusible compounds from a fungal host (*R. solani* AG5 and *A. alternata*) on the growth of the RNAi mutant strains was evaluated (Figs. 1D & E and S3 & 4). Interestingly, the $\Delta dcr2$ and the $\Delta dcr1\Delta dcr2$ double mutant showed a significant growth reduction compared to the WT (Figs. 1D-E). These results suggest that the Dcr2 component regulates genes involved in detoxifying diffusible metabolites.*

Q7. Line 117: Instead of "WT strain", please say "the *T. atroviride* WT strain", otherwise, it seems like "WT" refers to the AG2 and Ag5 fungi.

Line 177: change "it was harder to colonize" to something like "it struggled against *A. alternata*" or "it took longer to grow over *A. alternata*". The way it's currently phrased is awkward.

R7. We modified the text to read:

To evaluate the biocontrol capacity of the *T. atroviride* WT and RNAi mutant strains, we performed dual confrontation assays against *Alternaria alternata*, *Fusarium oxysporum*, *Fusarium* spp., *B. cinerea*, and two strains of *R. solani*, one belonging to anastomosis group 2 (AG2) and one to group 5 (AG5) (Table 1). Even though the WT strain overgrew all fungi (Fig. S1), it took longer to overgrow *A. alternata* than *R. solani* AG5, while it overgrew faster *R. solani* AG2 than AG5. In contrast, the $\Delta dcr2$ mutant and the double $\Delta dcr1\Delta dcr2$ mutant of *T. atroviride* showed reduced capacity to overgrow *A. alternata*, *R. solani* AG2, *R. solani* AG5, and *B. cinerea* (Figs. 1A and S1). We observed a clear zone of growth inhibition of the *Trichoderma* mutants by the fungal hosts (dashed rectangles in Figs. 1A and 1B). In contrast, the mutant strains in any RNA-dependent RNA polymerases ($\Delta rdr1$, $\Delta rdr2$, and $\Delta rdr3$) and those affected

in *ago1* and *ago2* overgrew the fungal hosts like the WT (Fig. S1). Interestingly, the $\Delta dcr2$ and $\Delta ago3$ strains did not stop the growth of *A. alternata* and *R. solani* AG5 and did not overgrow them (Fig. 1A & B).

Q8. Line 118: *dcr2* and double *dicer* mutant of what? *T. atroviride*? Please specify, in order for the reader to keep the fungal names clear.

Figure S1 needs to be clarified, similar to the figure legend for Figure 1A... what is the top row?

R8. We clarified the text (see previous response) and modified the legend to Figure S1 for clarity as follows:

Figure S1. Dual culture assays of RNAi mutant strains against some fungal hosts. *The T. atroviride* strains were inoculated on the left side, and the fungal hosts on the right side. The *T. atroviride* strains were inoculated on the left side, and the fungal hosts on the right side. The plates in the top row show the *T. atroviride* strains growing alone, and those in the right-most column represent the fungal hosts growing alone. Mycelium plugs of *T. atroviride* strains were inoculated on the left side of the Petri dish, and the fungal hosts were inoculated on the right side of the plate. The dual cultures were established in PDA at 27°C.

Q9. Line 122: keep the mutant names consistent... if you use a delta symbol in front of some of them, please do it with all of them. Here, the *ago* mutants have no delta symbols.

R9. We have now used the symbol delta in all cases when referring to the mutants.

Q10. Line 127: Here, I would mention briefly how you generated Figure 1C and S2. Perhaps just stating "a cellophane assay", or briefly stating what you describe in the MM section, is fine. Finally, do you mean Figure 1D here? I think 1C is the Rhizoc diffusibles, right?

R10. We have included a better description of the experiments in the results section, which now reads:

Due to the significance of antibiosis in biocontrol mediated by Trichoderma, we evaluated the impact of diffusible compounds released by the RNAi mutant strains on the growth of various fungal species (Figs. 1C and S2). Briefly, we grew the different T. atroviride strains on PDA plates covered by a cellophane sheet to allow the removal of Trichoderma and diffusion of metabolites and then inoculated the indicated fungal hosts. All mutant strains inhibited the growth of the different fungal hosts to a similar degree to that observed for the WT, except for

Δrdr2, which showed a strongly reduced capacity to inhibit the growth of all tested hosts (Fig. S2). Conversely, the effect of diffusible compounds from a fungal host (*R. solani* AG5 and *A. alternata*) on the growth of the RNAi mutant strains was evaluated (Figs. 1D & E and S3 & 4). Interestingly, the *Δdcr2* and the *Δdcr1Δdcr2* double mutant showed a significant growth reduction compared to the WT (Figs. 1D-E). These results suggest that the Dcr2 component regulates genes involved in detoxifying diffusible metabolites.

We also improved the description of the methods in the section “**Interaction in dual cultures**”.

Q11. Line 131-132: This statement seems a little bit over-reaching to me... I think this goes back to my earlier comment that in the introduction, it would be good to provide a bit of background on Dicer2 and its importance. I know it cannot all be covered, but just some general function statements. I think when you mess with Dicer2, there are a lot of indirect effects, and this could be one of them.

R11. We totally agree. As mentioned above, we incorporated a brief description of the role of each component of the RNAi machinery in the introduction. In addition, we modified the sentence to make clear that it is likely an indirect effect. The sentence now reads:

These results suggest that the Dcr2 component regulates genes involved in detoxifying diffusible metabolites.

Q12. Line 138: here, I really like that you use the term "host" to define the fungal plant pathogens. Maybe use "host" instead of prey, throughout.

R12. We decided to use the word host or fungal host throughout the manuscript.

Q13. Perhaps for figure 4, you could incorporate the "vs." (for versus) somewhere into that figure at the bottom? It almost reads like you are looking at ago and dicer mutants of *Rhizoc* and *Alternaria*, instead of looking at gene expression in *T. atroviride* versus these plant pathogens. It somehow should be made more clear, so people can look at the figure and understand exactly what is happening.

R13. We modified Figure 4 and added “vs” to avoid confusion.

Q14. Line 200: there is some italicized wording here that should not be italicized.

R14. We corrected it.

Q15. Line 206: please remind the reader what this phenotype was.

*Line 221-225, Figure 7: I understand that this is a very interesting finding.

However, I am wondering if the figure, as it stands, needs to be in the main body. It might be better as a supplementary figure, since in my reading of the manuscript, protein structure is not really fitting in with all of the gene expression analyses and plate assays that come before. It seems very interesting, yet disjointed at the same time. One thought is this - could you make Figure 6 supplementary, but pulling out the one graph for this particular NRPS, and then combine that graph with Figure 7?

R15. We strongly believe Figures 6 & 7 are very relevant for our manuscript. However, we had not described them well enough. We have reorganized Figures 6 & 7 and included an improved description of the findings in the results section, which now reads:

*Finally, due to the relevance of secondary metabolism in mycoparasitism (4, 14), we performed a prediction of biosynthetic gene clusters in *T. atroviride* to search for genes whose expression may be affected in the $\Delta dcr2$ strain. We identified core biosynthetic genes such as PKS and NRPS in 33 biosynthetic gene clusters (BGC). We observed evident defects in the expression of three PKSs and three NRPS belonging to three different BGCs in the $\Delta dcr2$ strain. These genes were expressed in the wild-type strain in all interactions but repressed in the $\Delta dcr2$ mutant (Fig. 6). Remarkably, within BGC 7.1, we found a gene (*Tatro_010797* gene) with a domain organization like that previously described in the NRPSs involved in the biosynthesis of the insecticidal hexa-depsipeptides destruxin, isaridin and isariin (Fig. 7A), all found in *Hypocreales* of the genus *Beauveria* (31). This gene is repressed in the $\Delta dcr2$ mutant in both mycoparasitic and control conditions (7B).*

Q16. Line 231: "*A. alternaria*" should be italicized.

R16. Thank you for pointing out this mistake. *A. alternata* was italicized.

Q17. Line 234: this sentence should have references, and more specifics... are these known effectors in *Trichoderma*? Other fungi?

R17. We rewrote the sentence to make it more specific and included references. The sentence now reads:

*These types of effectors have been found expressed in *Trichoderma* species in interaction with plants and mycoparasitism and are proposed to play a role in adhesion (32, 33).*

Q18. Line 244: change to "positively regulates"

R18. Modified as suggested.

Q19. Line 280-282: this sentence, as written, is unclear. Consider breaking it up into two sentences for clarity.

R19. We split the sentence into two sentences, as suggested.

Q20. Line 304: Are these from the BC, DC or AC conditions?

R20. We modified the sentence for clarity, which now reads:

*Based on the strong transcriptomic response of *T. atroviride* observed when confronted with *A. alternata* at all stages of the interaction (Figs. 3 and 4), we prepared small RNA-Seq libraries from *T. atroviride* mycelium obtained from the WT strain - *A. alternata* and $\Delta dcr2$ - *A. alternata* confrontations in the same conditions as the RNA-Seq libraries.*

Re: Spectrum03165-23R1 (Argonaute and Dicer are essential for communication between *Trichoderma atroviride* and fungal hosts during mycoparasitism)

Dear Dr. Alfredo H Herrera-Estrella:

Thank you for the privilege of reviewing your work. Below you will find my comments, instructions from the Spectrum editorial office, and the reviewer comments.

The manuscript is in principle accepted. However, there are still some suggestions from the reviewer #1 to improve clarity. Please, try to address them, and in the next revision round I will take the decision without submitting the manuscript back to the same reviewer. I apologize for all the delay in the decision process, but it was difficult to secure two reviewers in the beginning of the process. Thanks for your patience

Revision Guidelines

Sincerely,
Gustavo Goldman
Editor
Microbiology Spectrum

Reviewer #1 (Comments for the Author):

The manuscript is greatly improved over the initial version, and I appreciate all the effort the authors put into the resubmission. I

still find the manuscript to be complex with all the interactions, knockouts, and comparisons. It is only my opinion, but some additional help for the reader to keep track of the various comparisons could be beneficial, for example by color coding comparisons depending on the interactions tested between figures. A few additional minor points should be considered below.

Comments

1. Line 164-165, the authors mention that in BC and DC Cluster IX shows upregulation of genes involved in translation, but the cluster is only drawn over BC. Maybe the text can be slightly more carefully worded? I still don't really understand the colored bar on the left of Fig. 3A. The legend now explains that this is the different clusters identified in the dendrogram, but e.g., why are there multiple yellow regions? This is somehow separate from the clusters identified with Roman numerals?
2. Line 853, it might be helpful to say "T. atroviride WT" for clarity.
3. Line 238, should be "(Fig. 7B)" I believe.
4. Legend Figure S4, "t-student test" might refer to "Student's t test"? Is this done as a posttest of the ANOVA or separately? It would be appropriate to control for multiple comparisons here.
5. Line 358-360 has some repetition and should be revised.
6. Some of the text in Figure 10 is very small compared to all of the other figures.

Reviewer #2 (Comments for the Author):

All of the revisions have been carefully considered and carried out in a thoughtful manner. The manuscript is now more digestible and should be of interest to even a wider audience.

Dear Editor,

We thank the reviewers for their constructive comments, which have certainly helped improve our manuscript. Below, you will find a point-by-point rebuttal to the reviewer's comments.

Responses to Reviewer #1:

Comments

Q1. Line 164-165, the authors mention that in BC and DC Cluster IX shows upregulation of genes involved in translation, but the cluster is only drawn over BC. Maybe the text can be slightly more carefully worded? I still don't really understand the colored bar on the left of Fig. 3A. The legend now explains that this is the different clusters identified in the dendrogram, but e.g., why are there multiple yellow regions? This is somehow separate from the clusters identified with Roman numerals?

R1. The reviewer is correct in his observation regarding cluster IX. The text was wrong, and we have corrected it accordingly. The text now reads:

At the BC stage, T. atroviride showed an up-regulation of genes enriched in translation, RNA metabolism, and amino acid metabolism only with R. solani AG2 (Fig. 3, cluster IX).

We agree that the colored bar on the left of Fig.3A, rather than helping the reader, created confusion. The bar was linked to the clusters identified with Roman numbers. However, the numbers only indicated the main clusters and no subclusters, which appeared in different tones of the same color that were difficult to distinguish. Therefore, we opted to remove the colored bar.

Q2. Line 853, it might be helpful to say "T. atroviride WT" for clarity.

R2. Agreed, the text now reads:

(B) Venn diagram classifying DEGs, including an adjusted FDR q-value (FDR < 0.05) in the T. atroviride WT strain compared against the control condition. Before contact (BC), During contact (DC), and After contact (AC).

Q3. Line 238, should be "(Fig. 7B)" I believe.

R3. The reviewer is correct. We now mention Figure 7B as follows:

This gene is repressed in the $\Delta dcr2$ mutant in both mycoparasitic and control conditions (Fig. 7B).

Q4. Legend Figure S4, "t-student test" might refer to "Student's t test"? Is this done as a posttest of the ANOVA or separately? It would be appropriate to control for multiple comparisons here.

R4. Thank you for this observation. We modified Figure S4. The data were now analyzed using a one-way ANOVA and a TukeyHSD test, which is indicated in the new legend:

Figure S4. Growth inhibition exerted by *A. alternata* diffusible compounds on the *T. atroviride* strains. (A) The photographs show the effect of soluble compounds produced by *A. alternata* on the growth of the *T. atroviride* WT and RNAi machinery mutant strains. The first column corresponds to the growth of the *T. atroviride* strains on plates without diffusible compounds (control). The second column shows the growth of the *T. atroviride* strains on plates where *A. alternata* was previously grown (treatment). Growth of the *A. alternata* strain is shown at the top. (B) Colony diameter of the *T. atroviride* strains growing in the presence of *A. alternata* diffusible compounds (red bars, treatment) or without them (black bars, control). The values represent the mean of three biological replicas. A one-way ANOVA and a TukeyHSD test were performed to determine statistical differences among the strains in the same treatment ($P < 0.05$).

Q5. Line 358-360 has some repetition and should be revised.

R5. We agree with the reviewer and modified the text to read:

*To explore the potential cross-species regulation of *T. atroviride* small RNAs in the host fungus, we searched for target genes in the *A. alternata* genome. The analysis of target genes for the siRNAs induced in the WT strain at the DC and AC stages (Data set 2.12) uncovered 2,132 potential targets in *A. alternata* (Data set 2.12).*

Q6. Some of the text in Figure 10 is very small compared to all of the other figures.

R6. The reviewer is correct. We significantly increased the font size in the figure.

Re: Spectrum03165-23R2 (Argonaute and Dicer are essential for communication between *Trichoderma atroviride* and fungal hosts during mycoparasitism)

Dear Dr. Alfredo H Herrera-Estrella:

I apologize for the long delay to reach a decision. The manuscript is ready for publication. Congratulations !!!!!

Your manuscript has been accepted, and I am forwarding it to the ASM production staff for publication. Your paper will first be checked to make sure all elements meet the technical requirements. ASM staff will contact you if anything needs to be revised before copyediting and production can begin. Otherwise, you will be notified when your proofs are ready to be viewed.

Sincerely,
Gustavo Goldman
Editor
Microbiology Spectrum

Reviewer #1 (Comments for the Author):

The authors have addressed my concerns.